# Monocytic and granulocytic myeloid derived suppressor cells differentially regulate spatiotemporal tumour plasticity during metastatic cascade

Maria Ouzounova[1,*], Eunmi Lee[1,*], Raziye Piranlioglu[1], Abdeljabar El Andaloussi[1], Ravindra Kolhe[1], Mehmet F. Demirci[1], Daniela Marasco[2], Iskander Asm[1], Ahmed Chadli[1], Khaled A. Hassan[3], Muthusamy Thangaraju[1], Gang Zhou[1], Ali S. Arbab[1], John K. Cowell[1] & Hasan Korkaya[1]

It is widely accepted that dynamic and reversible tumour cell plasticity is required for metastasis, however, *in vivo* steps and molecular mechanisms are poorly elucidated. We demonstrate here that monocytic (mMDSC) and granulocytic (gMDSC) subsets of myeloid-derived suppressor cells infiltrate in the primary tumour and distant organs with different time kinetics and regulate spatiotemporal tumour plasticity. Using co-culture experiments and mouse transcriptome analyses in syngeneic mouse models, we provide evidence that tumour-infiltrated mMDSCs facilitate tumour cell dissemination from the primary site by inducing EMT/CSC phenotype. In contrast, pulmonary gMDSC infiltrates support the metastatic growth by reverting EMT/CSC phenotype and promoting tumour cell proliferation. Furthermore, lung-derived gMDSCs isolated from tumour-bearing animals enhance metastatic growth of already disseminated tumour cells. MDSC-induced 'metastatic gene signature' derived from murine syngeneic model predicts poor patient survival in the majority of human solid tumours. Thus spatiotemporal MDSC infiltration may have clinical implications in tumour progression.

[1] Department of Biochemistry and Molecular Biology, Georgia Cancer Center, Augusta University, Augusta, Georgia 30912, USA. [2] Department of Pharmacy, University of Naples Federico II, Naples 80138, Italy. [3] Department of Internal Medicine, Comprehensive Cancer Center, University of Michigan, Ann Arbor, Michigan 48109, USA. * These authors contributed equally to this work. Correspondence and requests for materials should be addressed to H.K. (email: hkorkaya@augusta.edu).

Metastatic disease is the end stage of extremely inefficient processes that entails overcoming multiple barriers. Evidences from preclinical and clinical settings suggest that dissemination of malignant cells is an early process[1]. However, majority of disseminated cells are either eliminated in circulation or remain dormant in distant organs including bone marrow, while very few cells eventually develop successful metastasis[1–3]. Therefore, the mechanism by which disseminated cells go on to establish successful metastasis is of utmost importance. S. Paget's 'seed and soil' hypothesis[4] for metastasis was a key milestone in cancer research that determined the direction of subsequent studies. Isaiah J. Fidler and others provided an unequivocal confirmation of the concept suggesting that some organs were more conducive than others for disseminated tumour cells 'seed' to grow[2,5,6]. Advanced studies in recent decades reframed the 'seed and soil' concept in a modern context by which successful metastases require that developing malignant cells eliminate anti-tumour responses, a small subset of (disseminating) cells -'seed'- undergo epithelial–mesenchymal transition (EMT) leading to cancer stem cell (CSC) phenotype and remotely generate a supportive microenvironment -'soil'- in distant tissues[7,8]. It is also accepted that successful colonization in distant organs requires disseminated tumours to revert back to epithelial phenotype via mesenchymal–epithelial transition (MET) to promote tumour cell proliferation[9]. Furthermore, a dynamic and reversible transitions between EMT and MET state has been shown to be critical processes in driving squamous cell carcinoma metastasis[9]. Consistent with this notion, EMT signature alone fails to predict metastasis in majority of malignancies[7,10,11].

Emerging evidences suggest that tumour-infiltrated immune cells (from mainly myeloid origin) differentiate into cells that promote tumour growth and invasion in addition to their immunosuppressive role[12,13]. Although myeloid-derived suppressor cells (MDSC) were initially identified in cancer patients and mouse models due to their potent immune-suppressive activity, they are now being implicated in the promotion of tumour metastasis by participating in the formation of pre-metastatic niches, angiogenesis and invasion[13]. MDSCs are heterogeneous population of immature myeloid cells that include monocytic (mMDSC) and granulocytic (gMDSC) subsets both of which have been shown to be immune-suppressive. Majority of studies do not distinguish between these two subsets, however, here we provide evidence that monocytic and granulocytic subsets not only have distinct molecular properties and distinct gene expression profiles but also have opposing effects on tumour cells. We show that 4T1 murine tumours in immune-competent mouse model develop spontaneous metastasis primarily to the lungs while the less invasive EMT6 tumours fail to generate any detectable metastasis. Furthermore, 4T1 tumours compared to the less invasive counterpart induced early induction and infiltration of mMDSCs in primary tumour and gMDSCs in the lungs. Using co-culture experiments, we show that tumour-infiltrated mMDSCs from 4T1 tumour-bearing mice induce EMT/CSC phenotype, while gMDSCs from lungs suppress EMT/CSC phenotype and promote cell proliferation. Furthermore, a 'metastatic gene signature' identified in a murine model predict poor patient survival human malignancies suggesting clinical relevance of our data in mouse models.

## Results

**Characterization of murine mammary tumours in syngeneic mice.** To investigate the role of immune system in the metastatic process, we used the metastatic (4T1) and less invasive (EMT6) murine mammary cell lines in a syngeneic (BALB/c) mouse xenograft model. Murine 4T1 cells were originally isolated from a spontaneous mammary tumour in the BALB/c strain and have been reported as metastatic and also exhibit the characteristics of human basal/triple-negative breast cancer (TNBC) subtype[14]. In contrast, the EMT6 and 67NR murine cell lines have been shown to be less invasive[15,16]. We first verified the tumorigenic and metastatic ability of EMT6 and 4T1 tumours, when 50,000 cells from both lines were injected into the mammary fat pads of, they produced similarly sized tumours within 8 weeks (Fig. 1a). The 4T1 tumours, however, showed pulmonary infiltrates as early as 1 week post implantation and developed spontaneous metastases by 5 weeks in 100% of animals (Fig. 1b), which also displayed enlarged spleens size and weight (Fig. 1c,d).

To determine whether the metastatic ability of the 4T1 murine tumours demonstrates both an epithelial–mesenchymal transition (EMT) phenotype and had cancer stem cell (CSC) properties, we used the CD29 and CD24 murine mammary stem cell markers as previously described[17]. As shown by immunofluorescence staining and flow cytometry analyses, 4T1 cells showed higher Vimentin expression compared to the EMT6 cells under serum-free culture conditions (Supplementary Fig. 1a,b) and also displayed a higher proportion of CSC as assessed by $CD29^+ CD24^+$ phenotype (Supplementary Fig. 1a–c).

It was previously demonstrated that the aggressive human basal–triple-negative breast cancer (TNBC) subtype produces higher levels of inflammatory cytokines compared to other subtypes[18,19]. To determine whether 4T1 murine tumour cells also secrete higher levels of inflammatory cytokines, we used a cytokine antibody array (Ray Biotech), which demonstrated that compared to non-metastatic EMT6 or 67NR cells, metastatic 4T1 tumours cells secrete higher levels of inflammatory cytokines/chemokines including IL6, IL8, RANTES, G-CSF, GM-CSF, IL12, CXCL16, CXCL5 and VCAM (Fig. 1e,f). It was previously demonstrated that SOCS3 negatively regulates inflammatory cytokines, G-CSF and GM-CSF in myeloid precursor differentiation[20–22]. Consistent with these data, 4T1 cells express lower level of SOCS3 protein compared to 67NR and EMT6 cells (Supplementary Fig. 1d) and that may account for the higher cytokine production in 4T1 cells.

Although the metastatic property of 4T1 tumour compared to EMT6 is well established in functional mouse transplantation studies, there was lack of detailed comparative gene expression analyses. We therefore performed mouse transcriptome analyses comparing the gene expression profile of 4T1 and EMT cell lines to their corresponding tumour xenografts grown in BLAB/c mice.

In comparison between 4T1 and EMT6 cells grown either *in vitro* or in orthotopic xenografts, >1,000 genes were identified that are differentially expressed between the *in vitro* cell lines versus xenografts (Supplementary Data 1 and 2) and 781 differentially expressed genes between 4T1 and EMT6 xenografts (Fig. 1g). Among these were genes implicated in metastasis and that were expressed at higher levels in 4T1 tumours (Fig. 1h). In contrast, upregulation of luminal cytokeratin, CK18 in 4T1 xenografts was not expected. However, CK18 is known to be up-regulated by number of stimuli in tumour microenvironment[23]. Interestingly the CSF3 gene (encoding G-CSF), a critical factor in accumulation of myeloid-derived suppressor cells (MDSC) was shown to be higher in 4T1 cell line as well as the 4T1 xenografts compared to the EMT6 counterparts. There were also several hundred common genes that were differentially regulated by both 4T1 and EMT6 xenografts (Supplementary Data 1). To further validate our findings, we chose AT-3 murine mammary tumour isolated from C57BL/6J mouse strain which was previously characterized as basal/TNBC with aggressive phenotype[24,25]. Cytokine antibody array showed that increased levels of the

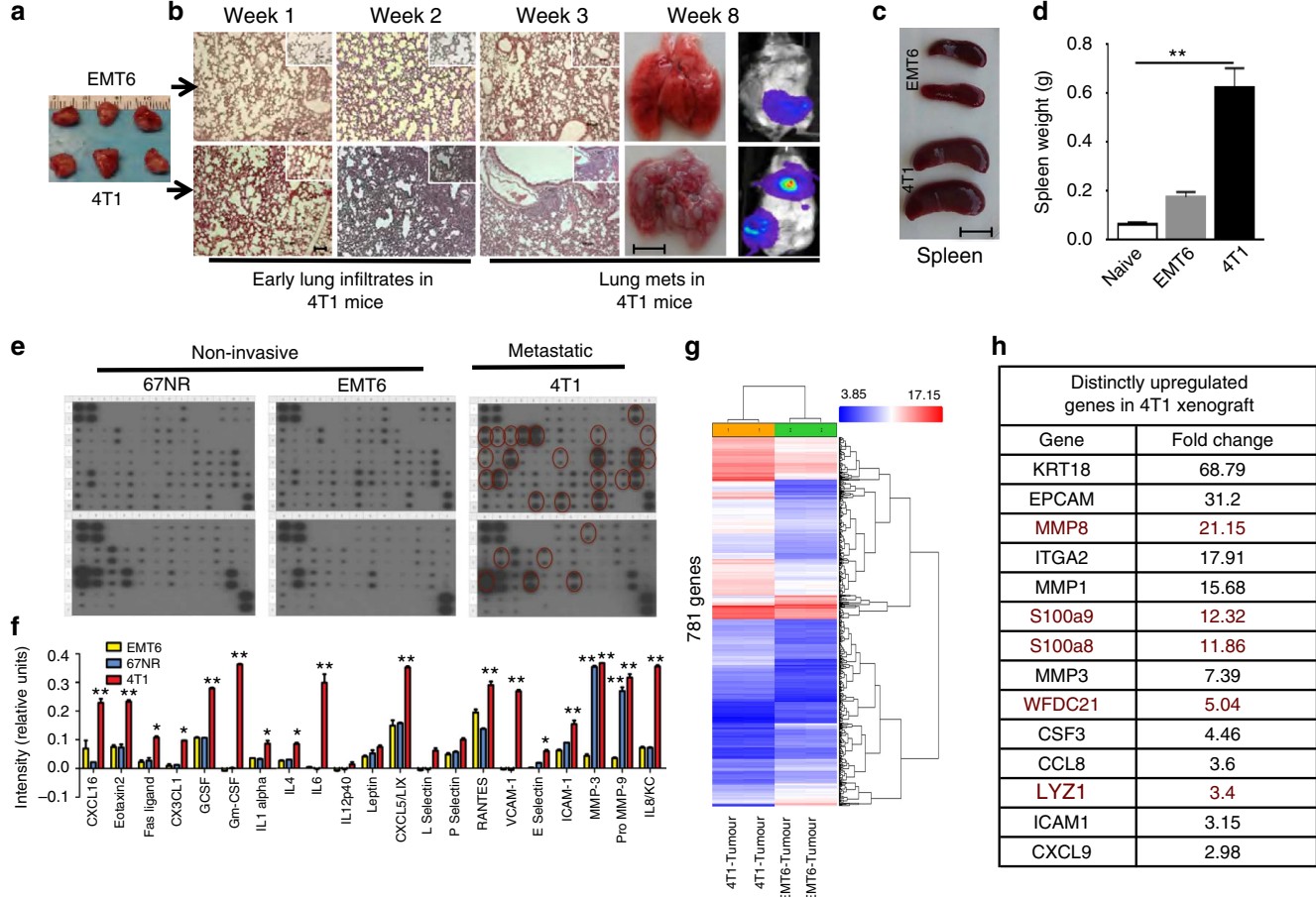

**Figure 1 | Biochemical and functional properties of metastatic 4T1 and non-invasive EMT6 murine tumours.** (**a**) Luciferase-expressing 4T1 and EMT6 murine mammary tumour cells generate same size tumours in syngeneic BALB/c mice. Scale bar, 1 cm. (**b–d**) 4T1 tumour-bearing animals develop spontaneous pulmonary metastasis and show early pulmonary infiltration and enlarged spleen compared to the EMT6 tumour-bearing animals. Scale bar, 100 μm (lung sections); and 1 cm (gross organ pictures). (**e,f**) 4T1 tumour cells produce higher levels of inflammatory cytokines compared to the non-metastatic EMT6 or 67NR as assessed by cytokine antibody array. (**g**) Mouse transcriptome analyses of 4T1 and EMT6 tumours grown in BALB/c mice reveal 781 differentially expressed genes. (**h**) Top common genes that are upregulated in both EMT6 and 4T1 tumours. (**i**) Select genes (implicated in metastatic process) that are distinctly upregulated in 4T1 tumour compared to the EMT6 are. Results are presented as mean ± s.d. ($n = 3$). *$P < 0.05$, **$P < 0.005$, unpaired $t$-test.

indicated cytokines including G-CSF and GM-CSF were detected in AT-3 cell line compared to the EMT6 (Supplementary Fig. 1e–g)

**Induction and infiltration of MDSCs by metastatic tumours**. MDSCs are heterogeneous population of immature myeloid cells with two main subsets; mMDSC and gMDSC. MDSC induction and infiltration in mouse mammary tumour models as well as the clinical setting of breast cancer have been previously reported[25,26,27]. However, the identity of the MDSC subsets in these tumours and their molecular mechanism of interaction with the tumour remain elusive. We investigated the systemic induction and infiltration of mMDSCs and gMDSCs in primary tumour, bone marrow, spleen and lungs at weeks 1–4 post implantation of EMT6 or 4T1 tumours. We demonstrate that early infiltration (as early as 1 week) of mMDSCs within primary tumours and a gradual increase of gMDSCs by week 4 was detected in the 4T1 tumours (Fig. 2a,b). Although pulmonary mMDSC infiltration was lower compared to infiltration in primary tumours of 4T1 tumour-bearing mice, pulmonary gMDSC infiltrates were increased >10-fold by week 3 (Fig. 2a,b) preceding the detection of metastatic lesions in the lungs (Fig. 1b). Flow cytometry analyses identifies, mMDSCs are characterized by

CD11b$^+$Ly6C$^{hi}$Ly6G$^-$ phenotype compared with the gMDSC subset have a CD11b$^+$Ly6C$^{low}$Ly6G$^+$ phenotype (Fig. 2c). We also verified our findings using AT-3 tumour model in the C57BL/6J mouse strain, though with different kinetics, demonstrating a similar induction and infiltration of MDSC subsets in BM, spleen, primary tumour and lungs with different time kinetics due to differences in tumour growth rate (Fig. 2d,e). To determine whether these MDSC subsets derived from either BM and tumour show T-cell suppression, we performed *in vitro* suppression assays in the presence or absence of the BM- or tumour-derived MDSC subsets from 4T1 tumour-bearing mice. As shown in Fig. 2f, tumour-derived mMDSCs showed higher levels of T-cell suppression *in vitro* that may be due to higher levels of nitric oxide synthase (*NOS2*) and arginase 1 (*ARG1*; Fig. 2g,h). These findings suggest that mMDSCs and gMDSCs infiltrate in the primary tumour and lungs, respectively, to promote metastasis in addition to suppressing anti-tumour immune responses.

**Induction of EMT/CSC by mMDSC at the invasive edge**. To investigate the effects of MDSC subsets on tumours, mMDSC and gMDSC subsets were independently flow sorted from the bone marrow of 4T1 tumour-bearing animals and co-cultured with the

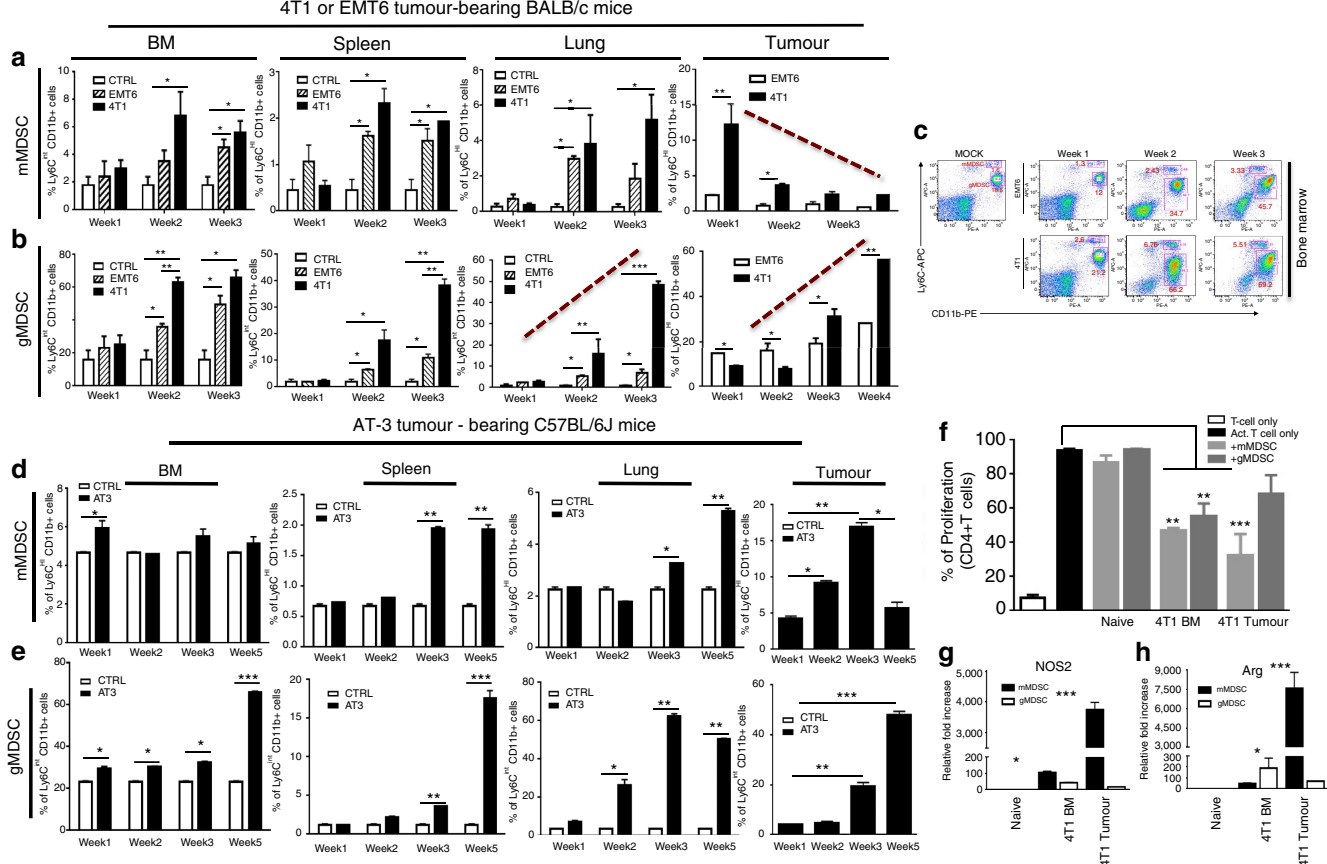

**Figure 2 | Tumour-induced expansion and infiltration of MDSC subsets in BALB/c or C56BL/6J mice bearing murine tumours.** (**a**,**b**) 4T1 metastatic tumour in BALB/c mice compared to the less invasive EMT6 tumour induce a systemic induction and infiltration of mMDSCs (CD11b$^+$Ly6C$^{hi}$) and gMDSCs (CD11b$^+$Ly6C$^{int}$) in bone marrow (BM), spleen, lung and tumour. (**c**) A representative flow cytometry data of MDSC subsets is shown. (**d**,**e**) AT-3 murine tumour in syngeneic C56BL/6J mice induces the expansion and infiltration of mMDSCs and gMDSCs in BM, lung and tumour in similar manner. (**f**) GM-CSF and G-CSF potently induce the expansion of MDSCs under *in vitro* culture condition. (**g**,**h**) Expression of NOS2 and Arg1 in MDSCs from 4T1 tumour-bearing mice are verified by qPCR Results are presented as mean ± s.d. ($n = 3$). *$P < 0.05$, **$P < 0.005$, ***$P < 0.0005$, unpaired $t$-test.

weakly metastatic EMT6 cells. As shown in Fig. 3a, mMDSCs but not gMDSCs show strong affinity towards the tumour cells and induce EMT phenotype (elongated spindle-shaped morphology), which was confirmed by strong Vimentin and CK14 expressions (Fig. 3b–e), established markers of EMT phenotype and invasion[28]. In contrast, gMDSCs failed to induce the expression of EMT markers (Fig. 3b–e). Analyses of EMT6 tumour-bearing animals also showed induction of MDSC subsets to a lesser degree, but they were less efficient in inducing the EMT phenotype (Supplementary Fig. 2a–d). Molecular studies confirmed that expression of Vimentin and Twist were only upregulated by mMDSCs derived from 4T1 tumour-bearing mice (Fig. 3f—red bars) compared with mMDSCs derived from EMT6 tumour-bearing animals (Fig. 3f—yellow bars).

To determine whether MDSCs from 4T1 tumour-bearing animals were more efficient in inducing cytokines compared with EMT6 tumour-bearing mice, we flow-sorted the MDSC subsets from primary tumours, bone marrow and lungs of 4T1 or EMT6 tumour-bearing mice and co-cultured them with EMT6 tumour cells. Using a multiplex assay, we determined that MDSCs from 4T1 tumour-bearing mice were more potent in inducing the levels of cytokines in co-culture experiments compared with those from the EMT6 tumour-bearing mice (Supplementary Fig. 2e,f—red and blue bars).

It is well established that increased expression of EMT markers correlate with enhanced invasive potential of tumour cells. Consistent with this notion, invasion of EMT6 cells was enhanced

when they were co-cultured with mMDSCs, while gMDSCs modestly (not significant) suppressed this process (Fig. 3g,h). We next reasoned whether induction of EMT by mMDSC also results in expansion of CSC population as previously reported[29]. EMT6 cells were co-cultured with mMDSCs from 4T1 tumour-bearing mice. EMT6 cells contain a small subset ($>1\%$) of CSC population as assessed by CD24$^+$CD29$^+$ phenotype[17] but when co-cultured with mMDSCs, there was more than threefold expansion of the CSC population; co-culture with gMDSCs had no effect on CSC levels (Fig. 3i). Expansion of the CSC population by mMDSCs was confirmed using the tumoursphere assay (Fig. 3j). To further corroborate the functional relevance of increased CSC population, we demonstrated that mMDSC-induced CSC population characterized by CD29$^+$CD24$^+$ phenotype showed higher tumorigenic potential compared to the differentiated CD24$-$CD29$^+$ phenotype (Supplementary Fig. 2g,h).

To confirm *in situ* localization of MDSC subsets, formalin-fixed, paraffin-embedded tumours (at week 1) and lung sections (at week 5) from 4T1 tumour-bearing mice were analysed using immunohistochemistry. Ly6C and Ly6G antibodies were used to identify mMDSCs and gMDSCs, respectively, Vimentin antibody as an EMT marker and Ki67 for proliferating cells. Ly6G positive cells were absent from the tumour at week 1, gMDSCs infiltrated in and around the metastatic lesions of lung at week 5 (Fig. 3k,l). Vimentin expression was restricted to the invasive edge in the primary tumour where mMDSC infiltration was seen (Fig. 3k).

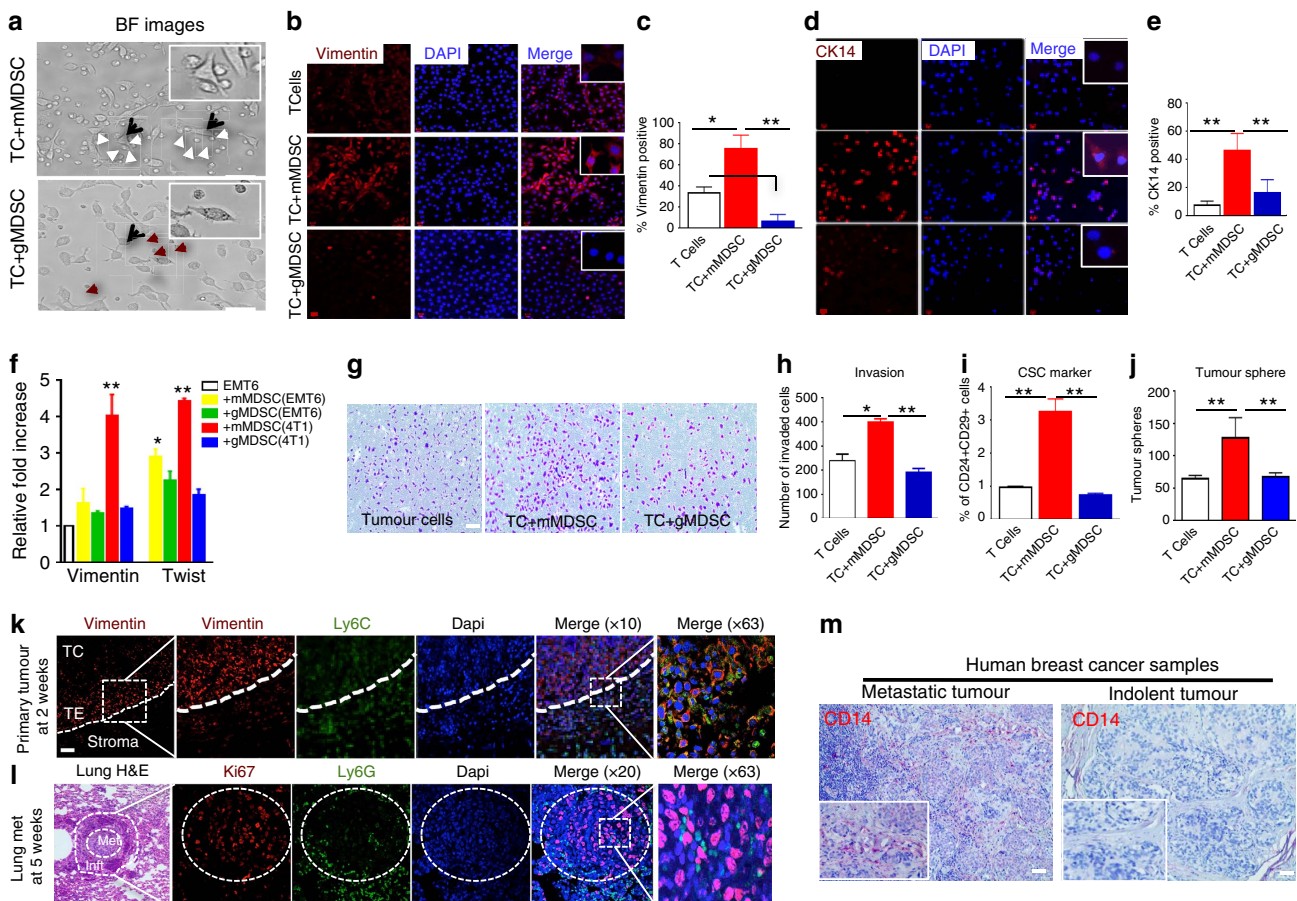

**Figure 3 | Monocytic-MDSCs localize at the invasive front of the primary tumour and induce epithelial–mesenchymal transition. (a)** Bright field (BF) images of co-cultures of EMT6 tumour cells (TC) (black arrow) with mMDSC (white arrows) or gMDSCs (red arrows) derived from 4T1 tumour-bearing mice. Morphological characteristic of EMT phenotype is visibly induced by the mMDSCs (top panel) which also show a strong affinity towards tumour cells. **(b–e)** mMDSC-induced EMT phenotype is assessed by enhanced expression of Vimentin and CK14. **(f)** Enhanced expressions of Vimentin and Twist induced by the mMDSCs from 4T1 tumour-bearing mice are verified by qPCR analyses. **(g–j)** mMDSCs from 4T1 tumour-bearing mice induce tumour cell invasion and CSC phenotype as shown by CD24-CD29 phenotype and tumour sphere forming assay. **(k,l)** *In situ* analyses of primary tumour and metastatic lesions by immunofluorescence staining reveal mMDSCs at the invasive edge of the primary tumour (TC, tumour center; TE, tumour edge) and gMDSCs around the pulmonary metastatic lesions in 4T1 tumour-bearing BALB/c mice (Met, metastasis; Inft, infiltrates). **(m)** Substantially higher number CD14-positive human mMDSCs were detected in metastatic human breast cancers compared to the indolent tumours. Results are presented as mean ± s.d. (n = 3). Scale bar, 50 μm; *P < 0.05, **P < 0.005; unpaired t-test.

These Vimentin-positive cells were Ki67 negative. However, in pulmonary metastatic lesions, the majority of tumour cells were Ki67 positive and co-localized with the gMDSCs (Fig. 3l).

Human mMDSCs are characterized by surface CD11b and CD14 expressions[30]. To provide evidence of mMDSC infiltration in human breast cancer samples, we performed immunohistochemical staining of 11 primary human tumour tissues with the CD14 antibody. There was higher levels of CD14-positive cells detected in metastatic tumours compared to the indolent tumours (Fig. 3m; Supplementary Fig. 3a,b).

**NOS2 production by mMDSCs induce tumour EMT/CSC phenotype.** To determine MDSC-induced global gene expression profile, we performed mouse transcriptome analyses. MDSC subsets derived from the primary tumour and BM of 4T1 tumour-bearing mice at 1 week post implantation were flow sorted by using Ly6C and CD11b surface antibodies. We performed the mouse transcriptome analyses either directly on these MDSC subsets or after co-culturing them with murine tumour cells *in vitro*. Results revealed that mMDSC and gMDSC subsets from BM or tumour displayed distinct gene expression profiles,

with over 1,000 differentially expressed genes (Supplementary Data 3). Moreover, mMDSCs show elevated expression of many EMT-related genes such as IL1a, IL6, TGFB1 and NOS2. In contrast, gMDSCs displayed expression of a different set of genes such as S100A8, S100A9, MMP8 and TGFb3 (Fig. 4a, Supplementary Data 3). To determine the effect of MDSC subsets on tumour cells, we then analysed the gene expression profiles of tumour cells that were co-cultured with mMDSC or gMDSC subsets derived from tumour or bone marrow of 4T1 tumour-bearing mice (Supplementary Data 4 and 5). Several hundred genes were differently expressed in EMT6 tumour cells when co-cultured with mMDSCs or gMDSCs (Fig. 4b). Tumour cells that are co-cultured with mMDSC showed more than twofold upregulation of EMT-related genes (Fig. 4b,c; Supplementary Data 4 and 5). Upregulation of these genes were confirmed by quantitative PCR (qPCR) using the indicated samples (Fig. 4d,e). Our findings from mouse transcriptome analyses were independently validated by qPCR in AT-3 tumour-bearing C57BL/6J mouse model, where the same genes were upregulated in tumour cells in response to the co-culture with mMDSCs derived from AT-3 tumour-bearing animals (Fig. 4f,g). To determine the effect

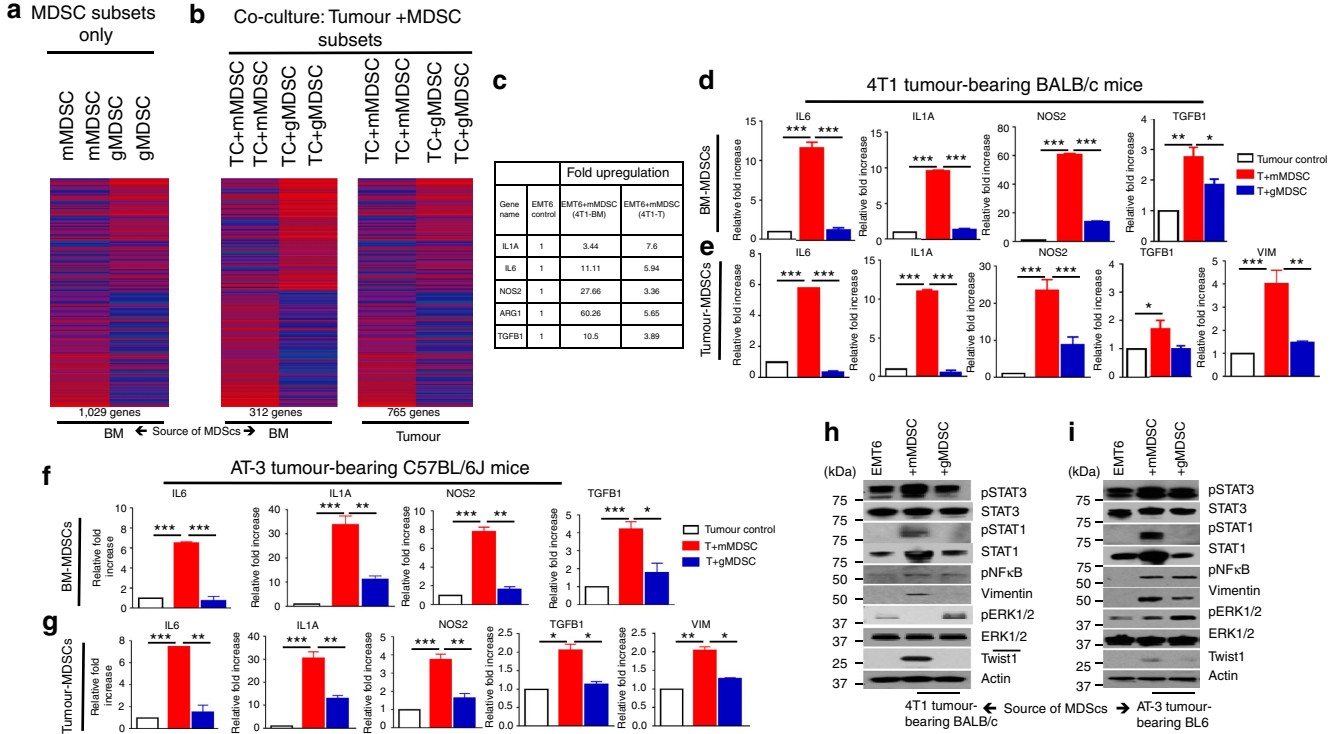

**Figure 4 | Monocytic-MDSCs promote tumour cell invasion and CSC phenotype via upregulation of EMT-related genes.** (**a**) Mouse transcriptome analyses of MDSC subsets from 4T1 tumour-bearing mice reveal a distinct gene expression profile between mMDSCs and gMDSCs and (**b**) induce distinct gene signature in tumour cells when co-cultured with tumour cells. (**c**) EMT-related genes are upregulated in tumour cells when co-cultured with BM or tumour-derived mMDSCs from 4T1 tumour-bearing BALB/c mice. (**d,e**) Upregulation of the indicated genes was verified by qPCR showing several fold induction by mMDSCs compared to the gMDSCs. (**f,g**) These genes were also upregulated by BM or tumour-derived mMDSCs from AT-3 tumour model in C57BL/6J mice. (**h,i**) mMDSCs from both 4T1/BALB/c and AT-3/C57BL/6J mouse model stimulate strong activation of pStat1 and pStat3, overexpression of Vimentin and Twist1 while suppressing pERK1/2 activation in tumour cells upon co-culture. Results are presented as mean ± s.d. (n = 3). *P < 0.05, **P < 0.005, ***P < 0.0005, unpaired t-test.

of MDSC subsets on the major signalling pathways, western blotting assay was performed on tumour cell lysates after overnight co-culture with mMDSCs or gMDSCs. Data support the gene expression analyses showing that MDSC subsets have differential effects on tumour cells. As shown in both 4T1/BALB/c and AT-3/C57BL/6J tumour models, mMDSCs induce a strong upregulation of pStat1, pStat3 and pNF-κB as well as enhanced expression of vimentin and twist in tumour cells, while suppressing the pERK1/2 activity (Fig. 4h,i). In contrast, gMDSCs show enhanced pERK1/2 activity while showing no effect on pStat1 and pStat3 activation (Fig. 4h,i). Enhanced pStat1 phosphorylation was also evident at the tumour invasive front where they were co-localized with Ly6C-positive mMDSCs (Supplementary Fig. 4).

We next sought to determine whether elevated NOS2 levels induce EMT/CSC phenotype in tumour cells. Treatment of EMT6 cells with NOS2 donor, DPTA induced the expressions of indicated genes; *IL1A*, IL6, *TGFB1* and *VIM* in dose-dependent manner as determined by qPCR analyses (Fig. 5a). In line with these findings, NOS2 inhibitor, 1400 W was able to suppress the mMDSC-induced transcription of these genes (Fig. 5b). Expectedly, NOS2 activation by DPTA induced the activation of pStat1 and pStat3 signalling pathways and protein levels of EMT markers, vimentin and twist, while the NOS2 inhibitor, 1,400 W suppressed the activation of the latter pathways and EMT markers (Fig. 5c). To assess the functional consequences of NOS2 mediated induction of EMT-related genes, we performed *in vitro* tumour invasion assay in presence and absence of NOS2 donor, DPTA. Treatment of tumour cells with DPTA resulted in

increased tumour cell invasion, however, blockade of NOS2 by 1400 W suppressed the mMDSC mediated tumour cell invasion (Supplementary Fig. 5a,b). Consistently, NOS2 activation also expanded the CSC population, while NOS2 blockade reduced the mMDSC-induced CSC population (Supplementary Fig. 5c,d). Together, these data suggest that mMDSCs may exert their effects on tumour cells via inducing NOS2 production.

**G-MDSCs promote primary and disseminated tumour cell growth.** We next examined the effect of MDSC subsets on tumour cell growth under *in vitro* co-culture conditions. Since 4T1 tumours develop spontaneous pulmonary metastasis in 100% of animals and show infiltration of gMDSCs in lungs 3 weeks post implantation, we reasoned that lung-infiltrated gMDSCs might support metastatic growth. To test this hypothesis, EMT6 tumour cells were co-cultured with mMDSC or gMDSCs from bone marrow (BM), tumour or lungs of 4T1 tumour-bearing mice. While tumour cell proliferation is enhanced by gMDSCs derived from lungs (60%) or tumour (40%), in contrast, bone-marrow-derived mMDSC or gMDSCs failed to do so (Fig. 6a,b). Lung-derived gMDSCs also enhanced the expression of EpCAM in tumour cells in co-culture experiments (Fig. 6c). We further confirmed that lung-derived gMDSCs from 4T1 tumour-bearing mice were more effective in promoting tumour cell proliferation compared to the gMDSCs from EMT6 tumour-bearing mice (Supplementary Fig. 6a,b).

Our mouse transcriptome analyses revealed that BM or lung-derived gMDSCs from 4T1 tumour-bearing animals regulate several hundred genes (757 and 764 genes, respectively) in

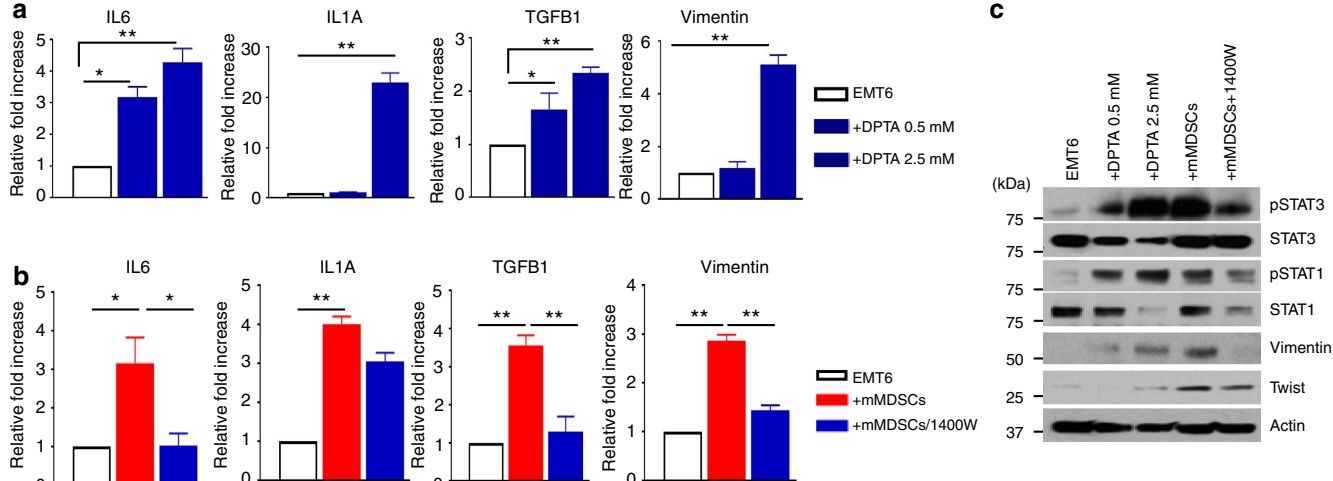

**Figure 5 | NOS2 production by MDSCs induce EMT/CSC phenotype.** (**a**) NOS2 induction by DPTA increased the expression of indicated cytokines and Vimentin in dose-dependent manner. (**b**) Inhibition of NOS2 by 1,400 W partially reduced the mMDSC-induced expression of IL1, IL6, TGFb and Vimentin. (**c**) NOS2 dependent pStat1 and pStat3 activation and EMT markers were confirmed by western blotting. Results are presented as mean ± s.d. ($n = 3$). *$P < 0.05$, **$P < 0.005$, unpaired t-test.

tumour cells upon co-culture (Fig. 6d,e; Supplementary Data 6 and 7). Interestingly, the top genes (S100A8, S100A9, MMP8, WFDC21, CCL3, LYZ1, FPR1 and TGFB2) that are upregulated up to several hundred fold upon co-culture with BM- or lung-derived gMDSCs (Fig. 6f; Supplementary Data 6 and 7). Surprisingly majority of these genes are also distinctly upregulated in 4T1 tumours compared to the cells grown under in vitro culture conditions (Fig. 1h). Upregulation of indicated genes as well as the proliferation marker PCNA were validated by qPCR analyses in both 4T1-BALB/c and AT-3/C57BL/6J tumour models (Fig. 6g–j).

**A murine gene expression signature predicts poor survival.** Although it is well established that EMT gene signature does not predict poor survival[7], ability of tumours to undergo dynamic EMT–MET transition is widely accepted to be a defining characteristic of metastatic cancers. Since the gMDSCs derived from metastatic 4T1 tumour-bearing animals were able to suppress the EMT-related genes and also induce distinct set of genes; S100A9, MMP8, S100A8, WFDC21, LYZ2, FPR1, CCL3, TGFb2 (Fig. 5f; Supplementary Data 6 and 7), we reasoned whether these genes could be specific for metastatic/aggressive behaviour. WFDC21 and Lyz2 are novel mouse genes, however, they are not identified in human genome yet. Therefore, we called remaining six genes (S100A9, MMP8, S100A8, FPR1, CCL3 and TGFB2) as 'metastatic gene signature' since they were particularly upregulated by lung-derived gMDSCs (Fig. 6f) and are also distinctly upregulated in 4T1 xenografts compared to the 4T1 cells grown under culture conditions (Fig. 1h).

We therefore tested the prognostic utility of the 'metastatic gene signature' in human samples using the TCGA data set (TCGA, Cell 2015)[31,32]. This signature predicted poor survival in breast cancer patients (Fig. 7a) and also correlated with enriched PCNA expression (Fig. 7b). Moreover, the metastatic gene signature correlated with higher expression of previously reported proliferation cluster genes[33] as 38 out of 40 genes were upregulated in patients with high metastatic signature (Fig. 7c). Although high PCNA expression alone predicts poor survival ($P = 0.0144$, Log-rank test) in breast cancer patients (Fig. 7d), combining PCNA and metastatic signature together very improves the poor survival prediction ($P = 1.018e − 4$,

Log-rank test) in the same cancer patients (Fig. 7e). Surprisingly, the metastatic gene signature identified in our mouse model-based studies was able to predict poor patient survival in 9 other solid tumours; brain lower-grade glioma, colorectal adenocarcinoma, renal cell carcinoma, cutaneous melanoma, endometrial carcinoma, ovarian serous adenocarcinoma, prostate adenocarcinoma and hepatocellular carcinoma (Fig. 7f). Altogether our data suggest that information from mouse models may have a clinical relevance.

**4T1 tumour-secreted cytokines enhance pulmonary metastasis.** We reasoned whether 4T1 tumour-secreted cytokines might enhance the metastatic ability of less invasive EMT6 tumours in mice. To provide evidence for this, we injected EMT6-Luci cells in BALB/c mice which were injected (intraperitoneally) with conditioned medium (CM) derived from 4T1 cells (Fig. 8a,b) which resulted in enhanced EMT6-Luci lung metastasis compared to control medium injected animals. Enhanced pulmonary metastasis also correlated with the expansion of MDSCs in 4T1 CM injected animals (Fig. 8c). To further corroborate, we generated 4T1-primed mice in which orthotopically injected 4T1 tumours were resected after 10 days and predicted that the 4T1 tumours within 10 days will create a pro-metastasis micro-environment. Therefore, the pro-metastatic microenvironment created by 4T1 tumours will enhance the metastatic ability of EMT6 tumours, which are otherwise non-metastatic. As predicted, the metastatic ability of EMT6-Luci tumours was enhanced in 4T1-primed mice (Fig. 8d, right panel) compared to the injection in naïve mice (Fig. 8d, left panel).

To determine the direct role of MDSC subsets in pulmonary metastasis, we injected 4T1-luci cells either alone or combined with lung-derived mMDSCs or gMDSCs isolated from 4T1 tumour-bearing mice. There was an enhanced pulmonary metastasis and shortened survival when 4T1-luci cells were injected in combination with gMDSCs compared to the 4T1-luci alone or combination with mMDSCs (Fig. 8e–h) supporting our findings that gMDSCs promote tumour growth. To further support these findings, we depleted gMDSCs in 4T1 tumour-bearing mice. Mice injected with 4T1 tumour cells (10,000 cells) via tail vein developed pulmonary metastasis (4 out of 5 mice) within 2–3 weeks (Fig. 9a), however, depletion of gMDSCs using

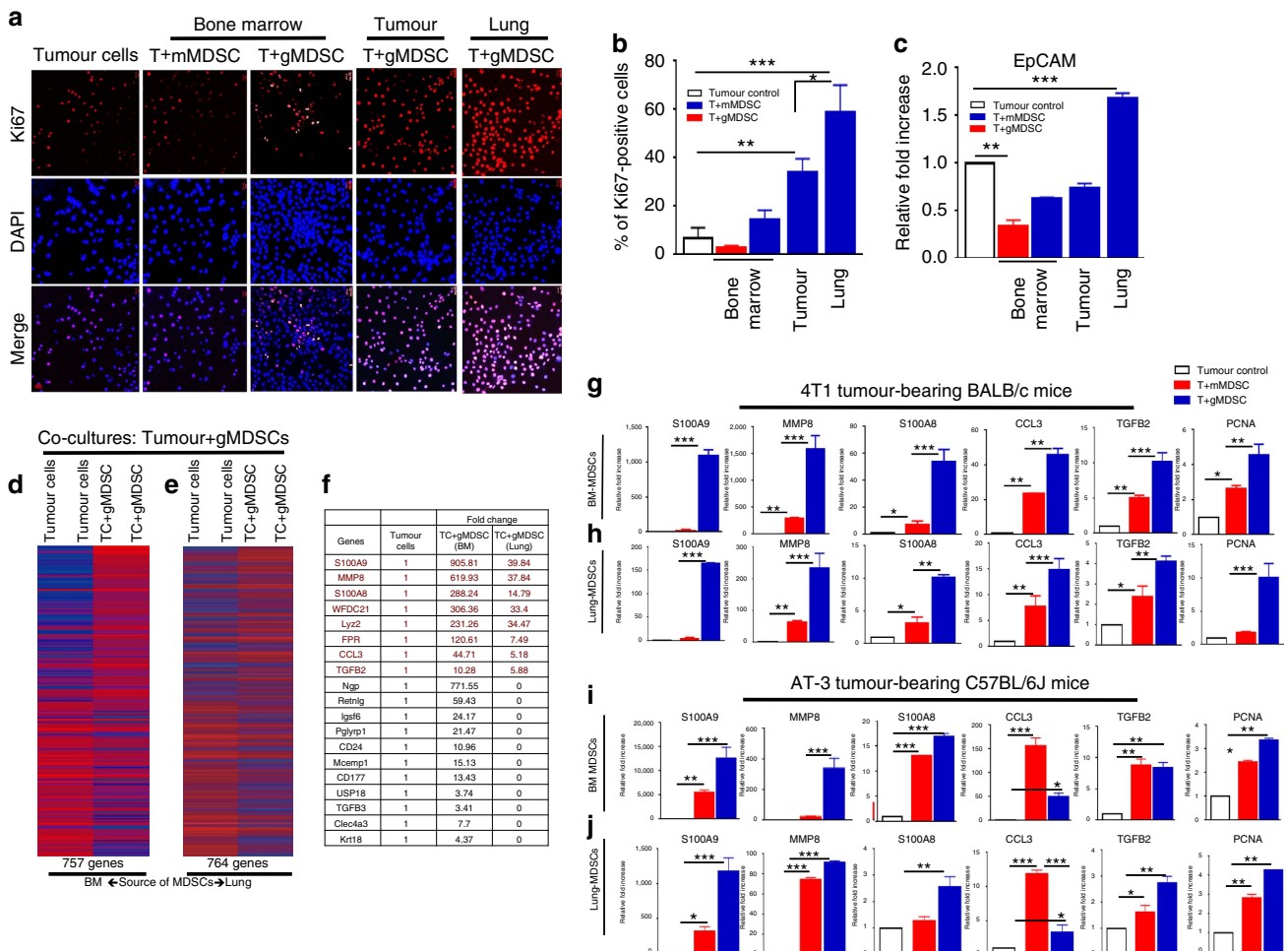

**Figure 6 | Lung-derived gMDSCs promote tumour cell proliferation via induction of a distinct gene expression signature called metastatic gene signature.** (**a,b**) Tumour cell proliferation was enhanced when they were co-cultured with lung- or tumour-derived gMDSCs compared to BM-derived mMDSCs and gMDSCs under serum-free culture condition. (**c**) There was also upregulation of EpCAM expression in tumour cells when they were co-cultured with lung-derived gMDSCs. (**d,e**) There are over 750 genes that are differentially regulated in tumour cells when co-cultured with gMDSCs either from BM or lungs of 4T1 tumour-bearing mice. (**f**) Top genes, highly upregulated in tumour cells upon co-culture with BM- or lung-derived gMDSCs are listed with their fold increase. We called 8 genes (indicated by red colour) 'metastatic gene signature' since they were distinctly upregulated by lung-derived gMDSCs. (**g–j**) Upregulations of these genes as well as the proliferation marker PCNA were verified by qPCR in both 4T1 tumour-bearing BALB/c and AT-3 tumour-bearing C57BL/6J mouse models. Results are presented as mean ± s.d. ($n = 3$). Scale bar, 50 μm; *$P < 0.05$, **$P < 0.005$, ***$P < 0.0005$ unpaired $t$-test.

anti-Ly6g antibody resulted in suppression of pulmonary metastasis (1 out 5mice; Fig. 9b,c) via reducing the infiltration of gMDSCs in the lungs (Fig. 9d). *Ex vivo* lung images clearly show the difference of metastatic growth between control and Ly6G-antibody treated groups (Fig. 9a,b,j bottom panels). We next examined the effect of gMDSC depletion in a spontaneous metastasis model where mice were treated with anti-Ly6g antibody after the orthotopic implantation of 4T1 tumour cells into the mammary fat pads. Treatment of mice with anti-Ly6g antibody reduced the tumour growth (Supplementary Fig. 7a–d) and pulmonary metastasis (Supplementary Fig. 7f). Our data also indicated that reduced gMDSC infiltration in tumour and lungs (Supplementary Fig. 7e,g) may account for reduced tumour growth and metastasis.

**Lung gMDSCs promote the growth of disseminated tumour cells.** To determine whether gMDSCs support the growth of already disseminated tumour cells, we developed a mouse model where orthotopically (fat pads) implanted tumours were resected at 1 week post implantation. Despite the presence of disseminated

tumour cells in regional lymph nodes and lungs, there was no metastatic growth up to 12 weeks of follow up (Fig. 10a–c). In contrast, majority of animals developed metastasis when the primary tumours were resected at 2 weeks post implantation (Fig. 10d,e). First of all, these findings provided further evidence that infiltration of gMDSCs in the secondary organs is required for successful metastasis. As shown in Fig. 2, expansion and infiltration of gMDSCs in lungs occur at 2 weeks post implantation. Second, this model may offer a great utility for investigation of the disseminated tumour cells in the absence of primary tumours as shown in the experimental outline (Fig. 10f). We utilized this model to evaluate the functional role of gMDSCs. In three groups of mice, luciferase-tagged primary tumours were resected at 1 week post implantation. First control group were not treated thereafter resection (Fig. 10g), second group were injected twice with tumour-derived mMDSCs (250 K per mice by tail vein; Fig. 8h) and third group were injected twice with lung-derived gMDSCs (250 K per mice by tail vein; Fig. 10i) isolated from 4T1 tumour-bearing animals. First and second group of mice were followed up for metastatic growth by bioluminescence imaging

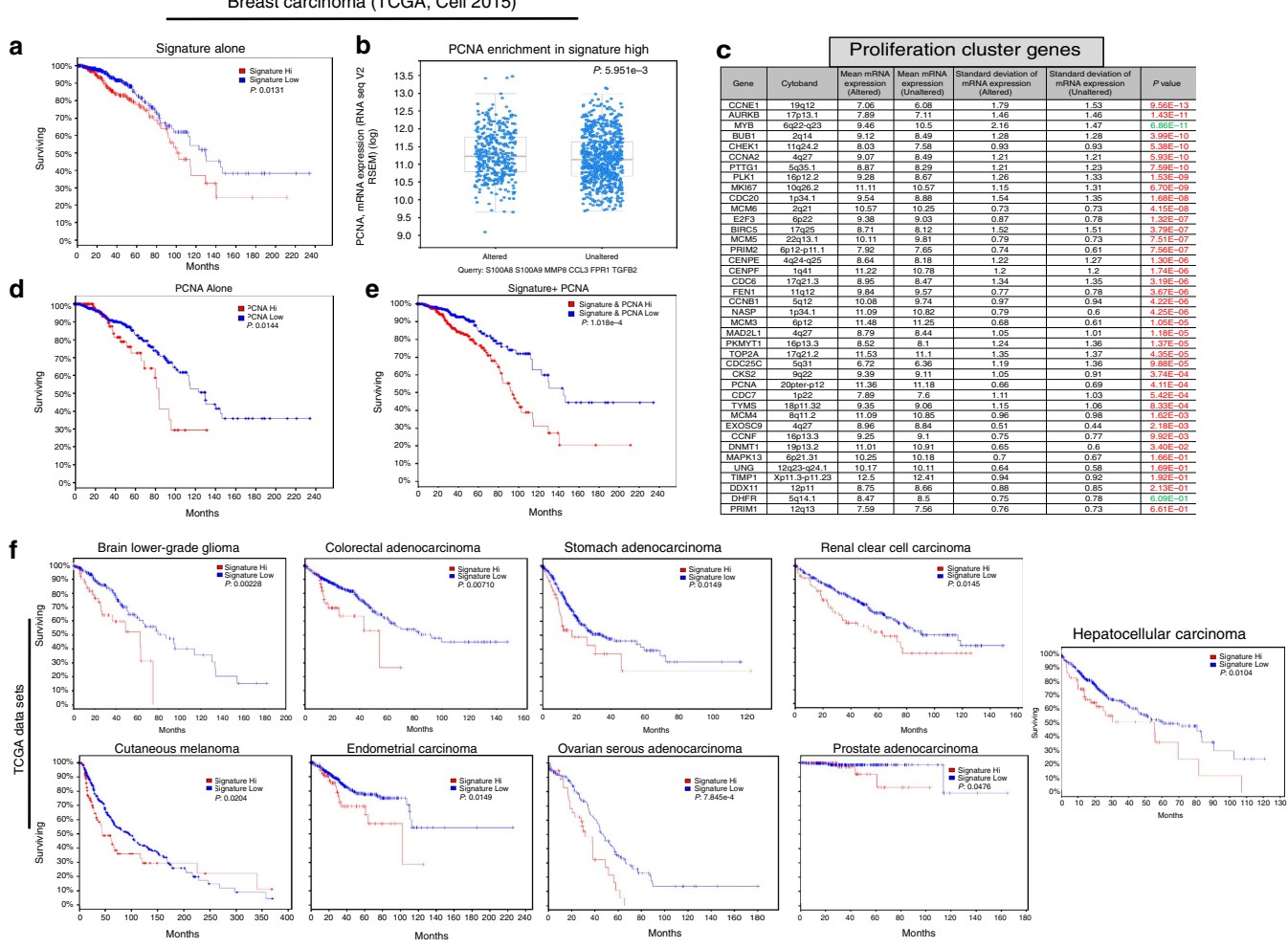

**Figure 7 | Mouse metastatic gene signature correlates with proliferation cluster genes and predicts poor survival in human malignancies.** (**a**) TCGA (2015, Cell) data set (971 patients) shows overexpression and/or amplification of corresponding human genes; S100A8, S100A9, MMP8, CCL3, FPR1, TGFb2 predict poor survival in breast cancer patients. (**b,c**) Tumours with high expression/amplification of these genes were also enriched in the expression of PCNA as well as the proliferation cluster genes. (**d,e**) PCNA overexpression alone predicts poor survival, however, survival prediction improves by several multitudes when combined with signature expression. (**f**) Overexpression/amplification of metastatic gene signature also predicts poor survival in majority of solid tumours that are indicated in the figure. Results are presented as mean ± s.d. P values (Log-rank test) are indicated in the figure.

(BLI) up to 11 weeks without any detectable metastasis (Fig. 10g,h). In contrast, 3 out of 4 mice injected with lung-derived gMDSCs developed metastasis (Fig. 8i,j). Collectively, our data suggest that dissemination and metastatic colonization/ growth are two independent steps in the metastatic cascade and may be regulated by different subsets of MDSCs as depicted by the illustration of our working hypothesis (Fig. 10k).

## Discussion

Our current understanding suggests that successful colonization in distant organs requires disseminated tumour cells with EMT phenotype to revert back to epithelial phenotype via MET. Despite the conceptual advances, microenvironmental cues and molecular crosstalk that regulate these dynamic phenotypic switches between EMT and MET in the primary site and distant organs are poorly elucidated.

Emerging studies implicated myeloid-derived suppressor cells (MDSC) in tumour progression and metastasis by facilitating the formation of pre-metastatic niches, angiogenesis and invasion[34]. However, MDSCs are heterogeneous population of the immature myeloid cells that include monocyctic and granulocytic subsets

both of which have been shown to be immune-suppressive[12]. Although the majority of studies do not distinguish between these two subsets[25,27], our studies provide first evidence that monocytic and granulocytic subsets not only have distinct molecular properties but also have opposing effects on tumour cells. We first show that both metastatic 4T1 and AT-3 murine breast tumour models of BALB/c and C57BL/6J mouse strains, respectively, induce early expansion and infiltrations of mMDSCs in the primary tumours where they facilitate dissemination of tumour cells at the invasive front by inducing EMT phenotype. In contrast, there were higher levels of gMDSC infiltrations (35–45%) in the lungs where these gMDSCs promoted the colonization and metastatic growth of disseminated tumour cells.

The mouse transcriptome analysis and qPCR assays of tumour-MDSC co-cultures, provided better understanding of the cross-communication. A strong affinity of mMDSCs towards tumour cells resulted in induction of EMT/CSC phenotype by upregulation of EMT gene expression signature while gMDSCs induced upregulation of factors that correlate with proliferation signature. Interestingly, number of EMT/CSC-related factors such

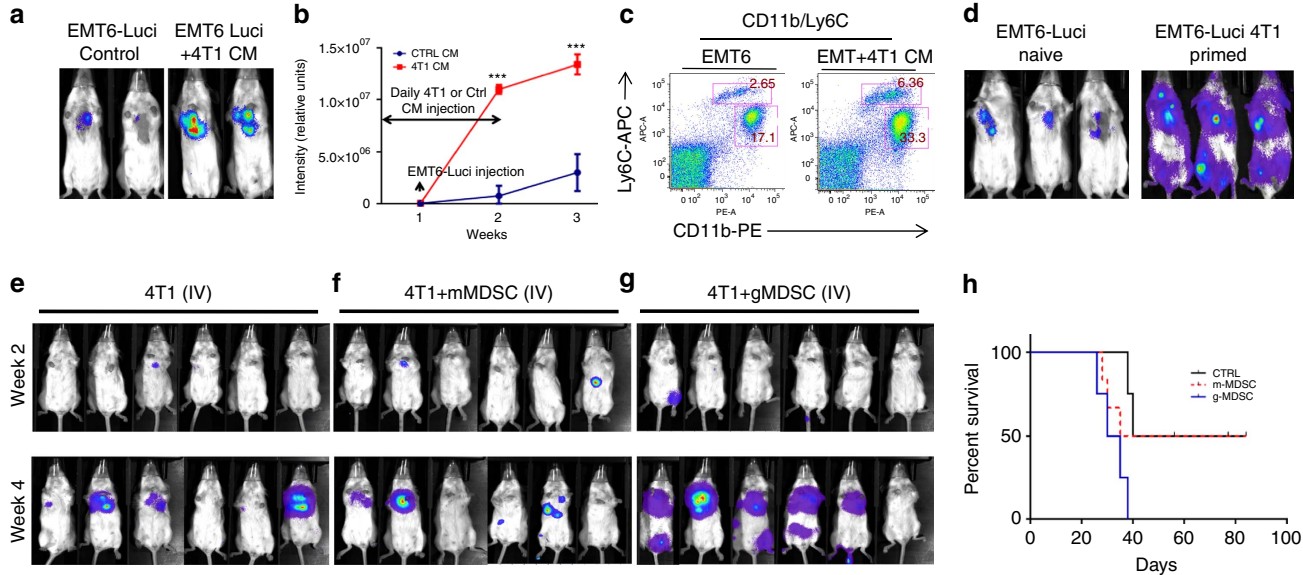

**Figure 8 | Metastatic growth of EMT6-Luci tumour is enhanced by 4T1 tumour-secreted factors that modulate MDSC induction and infiltration.** (**a–c**) EMT6-Luci cells were either IV injected alone or in combination with a condition medium from 4T1 cells that lead to enhanced metastatic growth and induction of MDSC subsets. ***$P < 0.0005$, two-way analysis of variance test (**d**) EMT6-Luci cells (50k per per injection) were either IV injected into naïve BALB/c mice or into the 4T1-primed mice (in which primary 4T1 tumours resected after 10 days) which resulted in enhanced metastatic growth. (**e–g**) Flow cytometry sorted tumour-derived mMDSCs (100k) or lung-derived gMDSCs (100k) from 4T1 tumour-bearing mice were co-injected with the 4T1-luci cells (50k per injection) and MDSC subsets injections were repeated 1 week later. (**g,h**) Animals injected with lung-derived gMDSCs showed accelerated metastatic growth and shortened survival compared to the control or mMDSCs co-injected group. Results are presented as mean ± s.d. (5–10 mice in each group).

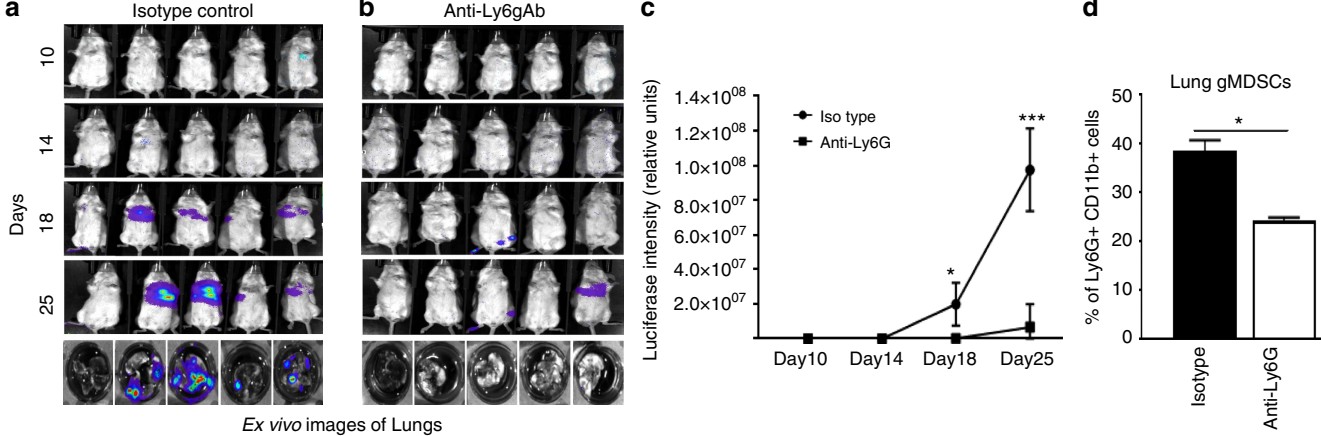

**Figure 9 | Depletion of gMDSCs suppress the pulmonary metastasis.** (**a–c**) Depletion of gMDSCs by anti-Ly6g antibody (200 µg/three times a week) in 4T1 tumour-injected mice resulted in suppression of pulmonary metastasis shown by optical imaging in life mice as well as *ex vivo* imaging of lungs. *$P < 0.05$, ***$P < 0.0005$, two-way analysis of variance test. (**d**) Reduced gMDSC infiltration in the lungs. Results are presented as mean ± s.d. (5–10 mice in each group). *$P < 0.05$, unpaired *t*-test.

as IL1a, IL6, NOS2 and TGFb1[29,35,36], were highly expressed in BM or tumour-derived mMDSCs and also further enhanced the expression of these genes in tumour cells upon co-culture. Activation of both Stat1 and Stat3 signalling pathways in tumour cells by mMDSCs may account for the induction of EMT/CSC phenotype as assessed by strong upregulation of EMT-related genes such as Vimentin, CK14 and Twist. In addition to number of inflammatory cytokines, NOS2 production in tumour cells was upregulated by mMDSCs. We demonstrated that mMDSC mediated expression of NOS2 is partially responsible for the induction EMT phenotype in tumour cells. Furthermore, mMDSCs were detected *in situ* at the invasive edge of the tumours where strong vimentin expression was observed.

Metastatic human breast cancer samples compared to the indolent tumours also showed higher infiltration of CD14-positive (human mMDSC marker) cells. Altogether these data provide a strong evidence that tumour-infiltrated mMDSCs induce EMT/CSC phenotype in primary tumours to facilitate dissemination. The morphological evidence of EMT and its requirement has been well established in preclinical and clinical studies[10,37]. However, recent studies provided evidence that EMT signature alone in breast cancer patients does not predict recurrence and disease-free survival[7,11].

Although dissemination via EMT conversion is a limiting necessary step in metastatic process, it does not warrant successful colonization and outgrowth in secondary organs[38]. In

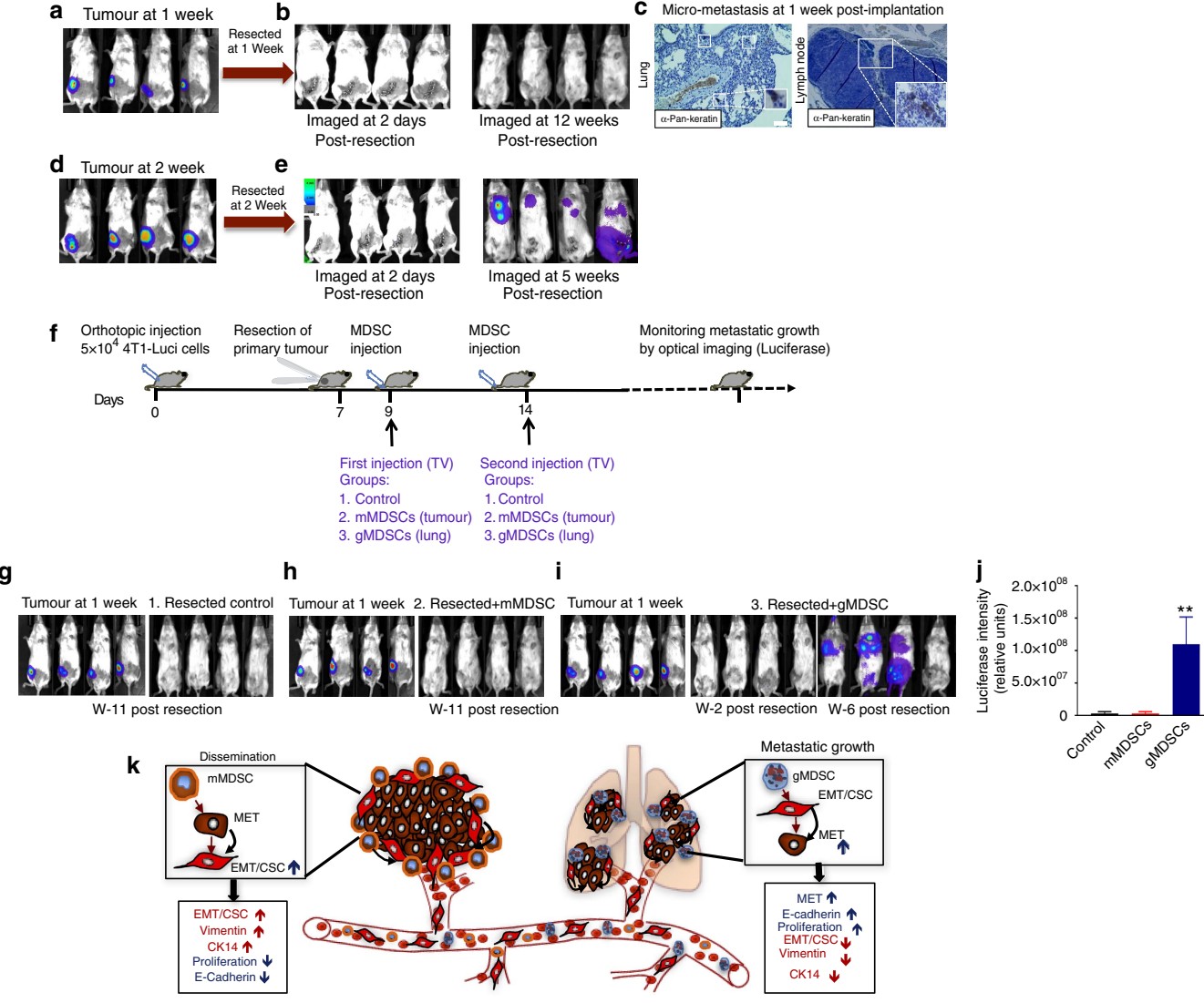

**Figure 10 | Lung-derived gMDSCs induce metastatic growth of disseminated tumour cells. (a,b)** We developed a mouse model that shows no metastatic growth when primary tumours were resected at 1 week post implantation **(c)** despite the existence of disseminated tumour cells in regional lymph nodes and lungs. **(d,e)** All mice develop metastasis when primary tumours were resected at 2 weeks post implantation. **(f)** Illustration of the experimental design. **(g)** Primary 4T1-Luci tumours were resected after 1 week post implantation and mice were followed up for metastatic growth by bioluminescence imaging (BLI). There was no metastatic growth up to week 11. **(h)** After resection of primary tumours, mice were injected (via tail vein) with tumour-derived mMDSCs as indicated and followed up by BLI without any metastatic growth. **(i,j)** Mice injected with lung-derived gMDSCs showed metastatic growth in three out of four mice. **(k)** Our findings suggest a spatiotemporal regulation of tumour plasticity by MDSC subsets in primary site and in distant organs as illustrated. Results are presented as mean ± s.d. (five mice in each group). Scale bar, 50 μm.

line with the notion, we provide evidence that massive pulmonary gMDSC infiltrates (35–45%) around 2–3 weeks post implantation may revert disseminated tumour cells back to epithelial phenotype and promote metastatic growth. Furthermore, our mouse transcriptome analysis of *in vitro* co-cultures and sample from syngeneic mouse model identified a 'metastatic gene signature' that predicts poor survival in majority of solid tumours including breast cancer. Interestingly, of the metastatic gene signature; S100A9, MMP8, S100A8 were also distinctly upregulated in the primary 4T1 xenografts in mice compared to its cell line grown in culture.

The fate of disseminated cells has been the area of intense investigation due to its clinical significance[39]. In our studies, we observed that total resection of primary 4T1 tumour 1 week post implantation eliminates the secondary metastatic growth despite the detection of disseminated cells in the regional lymph nodes

and lungs. We therefore called this as 'tumour dormancy model'. However, all animals develop metastatic growth when primary tumours were resected at 2 weeks post implantation. These data further provide evidence that pulmonary metastasis requires the infiltration of gMDSCs that begin after 2 weeks post implantation and thus resection of primary tumour at 1 week in the absence of gMDSC infiltrates result in tumour dormancy. We utilized the tumour dormancy model and demonstrated that adaptive transfer of lung-derived gMDSCs via tail vein injections supported the growth of already disseminated tumour cells. The requirement of gMDSCs in pulmonary metastasis was further validated by depleting gMDSCs using the Ly6g antibody in two different models of pulmonary metastasis.

In conclusion, we have provided convincing evidence for the spatiotemporal regulation of tumour plasticity by MDSC subsets in the primary site and distant organs using murine breast

tumours in syngeneic mouse model. We developed a novel tumour dormancy model and demonstrated the functional role of MDSCs in supporting the growth of already disseminated tumour cells. The 'metastatic gene signature' identified in murine models predicts poor survival in majority of human solid tumour patients including the women with breast cancer.

## Methods

**Cell lines and reagents.** 4T1, 67NR, E0771 and EMT6 cell lines were purchased from American Type Culture Collection (ATCC). All cell lines were analysed for mycoplasma contamination using MycoAlert Mycoplasma Detection Kit (Lonza). AT-3 cell line was a generous gift from Dr Kebin Liu (Augusta University). All cell lines were maintained in RPMI supplemented with 10% fetal bovine serum, and antibiotic/antimycotic 10,000 U ml$^{-1}$. CFSE (Invitrogen) staining was performed according to the manufacturer's instructions. Briefly, cells were washed twice in PBS, resuspended in PBS ($1 \times 107$ cells per ml) containing CFSE at a final concentration of 0.5 µM, and incubated for 10 min at room temperature. Subsequently cells were washed three times with saline and resuspended in complete medium at a concentration of $1 \times 10^6$ per ml. Recombinant mouse G-CSF (Shenandoah Biotechnology), SCF (Gemini Bioproducts), GM-CSF (BioLegend), IL6 (R&D Systems) were used at a final concentration of 50 ng ml$^{-1}$.

**Mouse studies and MDSCs depletion.** All mice procedures were conducted in accordance with the Institutional Animal Care and Use Committee at Augusta University. Balb/c female mice were purchased from Charles River at NCI and C57BL/6J female mice were purchased from The Jackson Laboratory.

4T1 murine breast cancer cells (50,000 cells per fat pad) expressing the luciferase gene were implanted into the fat pads of 5-week-old BALB/c or C57BL/J6 mice. For spontaneous metastasis model, primary tumours were resected 7 days after implantation and MDSCs were subsequently injected via the tail vein. For in vivo lung metastasis, 4T1-luc cells were mixed with MDSCs were injected into the tail vein. At least five animals per condition were used. For MDSC depletion studies, tumour cells were either othotopically implanted or injected via tail vein into the mice (five per group) and the one group of mice were treated with anti-Ly6g antibody (Bio X Cell) at 200 µg three times a week for the first 2 weeks and reduced to 100 µg. These mice were imaged utilizing the Caliper IVIS imaging systems.

**Protein and RNA analysis.** For western blotting analyses, cells were lysed in RIPA buffer (Sigma). 50 µg of each protein with Laemmli sample buffer were boiled for 5 min, and subjected to SDS–PAGE. The proteins were transferred onto PVDF membrane (Bio Rad Laboratories) using semi dry Trans-Blot (Bio Rad Laboratories). Blots were first incubated in TBS blocking buffer containing either 2% milk or 2% BSA (for phospho-specific antibodies) for 1 h at room temperature and then with the respective primary antibodies diluted in TBST (containing 0.1% Tween20 and 2% BSA) overnight at 4 °C. Subsequently, blots were washed and incubated with appropriate secondary antibodies (GE Healthcare) in TBST and detected using SuperSignal West Pico Chemiluminescent Substrate (Thermo). Antibodies to pSTAT3 (#9145), pSTAT1 (#8062), pNFkB (#3033), pErk1/2 (#9101), Erk1/2 (#9102) were purchased from Cell Signaling Technology and were used at 1:1,000 dilution. The antibodies to actin and vimentin were purchased from Santa Cruz Biotechnology. Uncropped scans of the most important western blots are provided (Supplementary Fig. 8).

Total RNA was extracted using RNeasy Mini kit (Qiagen) and 500 ng of RNA was used to make cDNA using iScript cDNA synthesis kit (Bio Rad).

For RT–PCR analyses, KiCqStart SYBR Green predesigned primers (Sigma) were used for the following genes: Vimentin (F—5′-GCCTGCAGGATGAGAT TCAGAATA-3′, R—5′-AACCAGAGGGAGTGAATCCAGATTA-3′), Twist (F—5′-CGGGTCATGGCTAACGTG-3′, R—5′-CAGCTTGCCATCTTGGAGTC-3′), Il6 (F—5′-AAGAAATGATGGATGCTACC-3′, R—5′-GAGTTTCTGTATCTCTCTG AAG-3′), Il1a (F—5′-CATAACCCATGATCTGGAAG-3′, R—5′-ATTCATGACAA ACTTCTGCC-3′), Nos2 (F—5′-CATCAACCAGTATTATGGCTC-3′, R—5′-TTT CCTTTGTTACAGCTTCC-3′), Tgfb1 (F—5′-CCCTATATTTGGAGCCTGGA-3′, R—5′-CTTGCGACCCACGTAGTAGA-3′), Tgfb2 (F—5′-GAGATTTGCAGGTATT GATGG-3′, R—5′-CAACAACATTAGGCAGGAGATG-3′), S100a8 (F—5′-ATACAA GGAAATCACCATGC-3′, R-, S100a9 (F—5′-CTTTAGCCTTGAGCAAGAAG-3′, R—5′-TCCTTCCTAGAGTATTGATGG-3′), Mmp8 (F—5′-AACTATGGATTCCCA AGGAG-3′, R—5′-CTTTGATTGTCATATCTCCAGC0-3′, Ccl3 (F—5′-CGGAAGA TTCCACGCCAATTC-3′, R—5′-GGTGAGGAACGTGTCCTGAAG-3′), and Pcna (F—5′-CTGAGGTACCTGAACTTTTTC-3′, R—5′-TATACTCTACAACAAGG GGC-3′). The relative expression mRNA level was normalized against the internal control GAPDH (F—5′-AAGGTCATCCCAGAGCTGAA-3′, R—5′-CTGCTTC ACCACCTTCTTGA-3′) or ACTB (F—5′-GATGTATGAAGGCTTTGGTC-3′, R—5′-TGTGCACTTTTATTGGTCTC-3′) gene (ΔCt = Ct (target gene) − Ct (internal control gene)). The relative fold change was measured by 2-ΔΔCt formula compared to the control cells. Means and differences of the means with 95% confidence intervals were obtained using GraphPad Prism (GraphPad Software Inc.).

Two-tailed Student's t-test was used for unpaired analysis comparing average expression between conditions. $P$ values $< 0.05$ were considered statistically significant.

**Gene expression analysis.** The RNA extracts were first analysed by a Nanodrop 2000 spectrophotometer (Thermo Fisher Scientific, Waltham, MA). RNA quality was determined by the ratios of A260/A280 (close to 2) and A260/A230 (close to 2). Qualified RNAs were further tested using an Agilent 2100 Bioanalyzer (Agilent Technologies, Santa Clara, CA), and samples with RIN > 7 were selected for microarray analysis using the Affymetrix MTA 1.0 (Affymetrix). The labelling, hybridization, scanning and data extraction of microarray were performed according to the recommended Affymetrix protocols. Briefly, the fluorescence signals of the microarray were scanned and saved as DAT image files. The AGCC software (Affymetrix GeneChip Command Console) transformed DAT files into CEL files to change image signals into digital signals, which recorded the fluorescence density of probes. Next, we used Affymetrix Expression Console software to pretreat CEL files through Robust Multichip Analysis (RMA) algorithm, including background correction, probeset signal integration, and quantile normalization. After pretreatment, the obtained chp files were analysed by Affymetrix Transcriptome Analysis Console software to detect differentially expressed genes.

**FACS and co-culture experiments.** To analyse MDSCs, single-cell suspensions were prepared from bone marrows, spleens, lungs and tumour tissues. Bone marrow cells were obtained by flushing bones with PBS using a 28G 1/2 syringe. Tumour and lung tissues were dissociated and digested with collagenase (Stem Cell Technologies) for 1hr at 37 °C. Red blood cells were lysed by ACK lysis buffer (Gibco). These cells were labelled with fluorescence-conjugated Ly6C (#128015-dilution 1/400), Ly6G (#127605-dilution 1/100) and CD11b (#101208-Dilution 1/200) antibodies (Biolegend) and analysed on a FACS canto flow cytometer (BD Biosciences). Different subsets of MDSCs were sorted with a FACS Aria cell sorter (BD Biosciences). For co-culture experiments, tumour cells were cultured alone (control) or with 1 µM carboxyfluorescein diacetate succinimidyl ester (CFSE, Molecular probes) labelled MDSCs in 10% FBS RPMI media unless indicated otherwise for 24–48 h at the ratio of 1:1. After incubation, FITC negative tumour cells were isolated with a FACS Aria cell sorter for following experiments. Culture supernatant were collected for cytokine analysis. Sterile coverslips were placed in dishes for immunofluorescence staining.

**Invasion assays.** Tumour cells ($5 \times 10^4$) after the co-culture with MDSCs were seeded into the top chamber of transwell inserts coated with Matrigel (BD). The inserts were placed in a 24-well plate containing culture media of 5% FBS RPMI. Invaded cells were counted after 16–18 h. Experiments were done in triplicates.

**In vitro suppression assay.** Splenocytes from naïve Balb/c mice were labelled with 1 µM CFSE and placed in triplicates into a U-bottom 96-well plates ($1 \times 10^5$) with or without the presence of 1 µg ml$^{-1}$ anti-CD3 Ab (#553057, BD Biosciences) and 5 µg ml$^{-1}$ anti-CD28 Ab (#557393, BD Biosciences). Labelled Splenocytes were co-cultured with purified MDSCs for 72 h and CFSE dilution of CD4$^+$ T-cell fractions was analysed by flow cytometry.

**Immunostaining.** For immunohistochemistry, paraffin-embedded sections were de-paraffinized in xylene and rehydrated in graded alcohol. Antigen retrieval was done by incubating the sections in citrate buffer pH 6 (Invitrogen) by either boiling for 10 min or in the microwave. Staining was performed using peroxidase HistostainPlus Kit (Invitrogen) according to the manufacturer's protocol. For fluorescent staining, cells were fixed with 4% paraformaldehyde at for 10 min. After rehydrating in PBS, cells were incubated with primary antibodies at room temperature for an hour, washed and incubated 30 min with fluorescence-conjugated secondary antibodies. The nuclei were stained with DAPI/antifade (Invitrogen) and coverslipped. Sections were examined with a fluorescent microscope (Leica). The following primary antibodies were used: Ly6C (#ab15627-Dilution 1/100, Abcam), Ly6G (#MAB1037-Dilution 1/100, R&D biosystem), Ki67 (#12202-Dilution 1/100, Cell Signaling), Vimentin (#550513 Dilution 1/100, BD Biosciences), CD14 (#10073-H08H Dilution 1/100, Sino biological). The secondary antibodies were purchased from Invitrogen.

**Tumour sphere assay.** Tumour tissue was dissociated mechanically and enzymatically using Collagenase/Hyalurinidase (Stem Cell Technologies). Cells were sieved sequentially through a 100-µm and a 40-µm cell strainer (Falcon) to obtain a single-cell suspension. Dissociated single tumour cells were plated on 6-well ultra-low attachment plates (Corning Inc.) at a density of $1 \times 10^5$ cells per ml and grown for 7–10 days. Subsequent cultures after dissociation of primary spheres were plated on new plates at a density of $1 \times 10^4$ cells per ml. Tumour sphere cultures were grown in a serum-free mammary epithelium basal medium[40].

**Cytokine antibody array.** Mouse Cytokine Array C1000 (RayBiotech) was used for detection.Cells were plated at equal number and cultured for 48 h. Conditioned media were collected and processed according to the manufacturer's

recommendation. Briefly, the membranes were blocked by incubation with the blocking buffer at room temperature for 30 min and incubated with the sample at 4 °C overnight. Membranes were washed 3 times with Wash Buffer I and 2 times with Wash Buffer II at room temperature for 5 min per wash and incubated with biotin-conjugated antibodies at room temperature for 2 h. Finally, the membranes were washed and incubated with horseradish peroxidase-conjugated streptavidin at room temperature for 2 h and with detection buffer for 2 min.

We used a luminescence detector (LAS-1000, Fujifilm) for detection, and the data were digitized and subjected to image analysis (ImageJ). By subtracting the background staining and normalizing to the positive controls on the same membrane, we obtained relative protein concentrations.

**Data availability.** The data discussed in this publication have been deposited in NCBI's Gene expression Ominbus under the GEO Series accession code GSE81701 (https://www.ncbi.nim.nih.gov/geo/query/acc.cgi?acc=GSE81701).

The TCGA data referenced during the study are in part based upon the data generated by the TCGA Research Network: http://cancergenome.nih.gov/ and are available in a public repository from the cBIoportal for Cancer Genomics website http://www.cbioportal.org/. All the other data supporting the findings of this study are available within the article and its Supplementary Information files and from the corresponding author upon reasonable request.

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

## Acknowledgements

We thank the core facilities; Georgia Cancer Center flow cytometry, integrated genomics, core imaging facility for small animals and histology for their excellent service and Dr Jeane Da Silva for her help in flow cytometry analyses and sorting. This work is supported by the Georgia Cancer Center start up fund and Komen Career Catalyst Research grant to H.K.

## Author contributions

H.K. conceived and designed the experiments, H.K., M.O., E.L., A.E.A., R.P., I.A. and M.F.D. performed the experiments, H.K., M.O. and E.L. analysed and interpreted the data, H.K., M.O., E.L. and J.K.C. contributed to preparing the manuscript, R.K., M.T., G.Z., A.S.A., K.A.H., D.M., A.C. and J.C.K. provided reagents.

## Additional information

**Competing interests:** The authors declare no competing financial interests.

