## [Peer Review File · Nature Communications]

Reviewers' comments:

Reviewer #1 (Remarks to the Author):

This is interesting study describing the role of gMDSC in tumor metastases in mouse model of breast cancer. Most of the experiments are well designed and conclusions are convincing.

Major concern is with rather large amount of data that are mostly confirmatory of already established in many publications facts. It includes detailed description of changes in MDSC in 4T1 model, the role of mMDSC in EMT (including STAT3 signaling), the role of gMDSC in promoting tumor metastases. There are new data concerning changes in tumor cells caused by gMDSC and elegant experiments in Figure 8.

Authors stated that EMT signature was not associated with tumor prognosis and referenced one paper (page 9). However, brief analysis of the literature showed more than two dozen papers reported just opposite conclusions. In this respect data in Figure 6 look rather incrementally different from already published data.

Specific concerns

1. Authors did not established causal role of gMDSC in the observed phenomenon by depleting these cells;
2. The authors check the cytokine secretion profile in co-culture of MDSC with tumor cells. It was not clear how they determined what cytokines were secreted only by tumor cells and not by MDSC (figure S2; they normalize data to secretion profile or tumor cells alone, but not info about secretion from MDSC alone).
3. In co-culture the authors do not mention GM-CSF and there is no indication of MDSC viability (and/or phenotype) after the co-culture.
4. The authors use immunocompetent mice (BalbC and B6), but they don't mention or discuss how they control the effect of endogenous neutrophils/MDSC during tumor development and when they inject exogenous MDSC in TBM.

Minor comments

1. the authors should quantify luciferase signal in mice and show statistical significance (some mice have very low signal that could be just background), also few mice are shown, so it is important to understand what is the frequency of the observed results.
2. the authors should show that after induction tumors keeps growing (luciferase detection and quantification)
3. The differences in signaling (Figure 4m-n) aren't discussed, or validated (for example, inhibit the signal with exogenous inhibitors gives the same results??). Also total stat 1 and stat 3 are missing in the WB.

Reviewer #2 (Remarks to the Author):

This is an elegant manuscript demonstrating a dynamic and reversible tumor cell plasticity involving epithelial to mesenchymal transition (EMT) and mesenchymal to epithelial transition (MET), which are hallmarks of metastatic dissemination and colonization. Whereas metastatic spread involves EMT leading to invasion, colonization at distant sites requires MET, a process conducive metastatic cell proliferation. Using co-culture experiments in vitro and in vivo, the authors demonstrate a critical interaction between mammary tumor cells and infiltrating monocytic and granulocytic subsets of myeloid derived suppressor cells (mMDSC, gMDSC) at the primary tumor and distant organs. They show that mMDSC stimulate EMT at primary tumor and that gMDSC stimulate MET at the lungs. A "metastatic gene signature" was identified that was further shown to predict poor patient survival in several human cancers, suggesting clinical relevance of

these studies.

This study uses multiple and alternative approaches to demonstrate the cross-communication between mMDSC and tumor cells in inducing EMT in tumors, leading to stemness and invasion. They also clearly show that infiltration of gMDSC in the lungs stimulate MET, which is conducive of tumor cell proliferation leading to lung colonization.

There remains however several issue to be addressed as outlined below.

1. This study revolves around tumor infiltrated immune cells or myeloid cells that differentiate into cells that promote tumor growth and invasion in addition to their immunosuppressive role. They show that mMDSC and gMDSC are immunosuppressive, however, it is not clear how the immunosuppressive activity is contributing to the diverse metastatic inducing properties.

2. "4T1 tumors showed pulmonary infiltrates as early as one week post implantation and also displayed enlarged spleens". It is not clear which immune cells are causing spleen enlargement and how these are related to the immunosuppressive mMDSC or gMDSC (Fig. 1c).

3. While infiltration of mMDSC peaks after one week in 4T1 primary tumor following implantation, it continues to infiltrate the lungs up to 3 weeks (Fig. 2a) suggesting mMDSC may play a role in lung colonization, although the authors show that they do not in co-culture experiments with the non-metastatic EMT6 cells. Also, it appears that the infiltration of mMDSC in AT-3 tumors (Fig. 2g) does not follow the same kinetics as for 4T1. What is the basis for this discrepancy?

4. Fig. 3g-h shows upregulation of vimentin and twist (EMT) upregulation in EMT6 co-cultured with mMDSC and upregulation of Ki67 (MET) in cells co-cultured with gMDSC. Is Twist expressed in vimentin positive cells and not in ki67+ cells?

5. Fig. 4 demonstrates that mMDSC infiltrating 4T1 tumors cause CSC expansion after co-culturing with EMT6 cells using the stem cell markers CD24+CD29+. Can this population be isolated and tested for tumor initiating potential and metastatic potential?

6. The authors show changes that indicate enrichment of CSCs as indicated above. They show increases in p-STAT and NFkb when tumor cells are incubated with mMDSC and increase in p-Erk when tumor cells are co-cultured with gMDSC (Fig. 4m-n). Although implicit in the EMT and MET phenotypes, these markers remain correlative. Is p-ERK expression more prominent at sites of gMDSC infiltration in the lungs and p-STAT at sites of EMT/invasion or at the leading edge. Is p-Erk associated with ki67+ cells (Fig 5a)?

7. 4T1 cells are thought to be similar to basal or triple negative breast cancer cells, yet 4T1 xenografts upregulate keratin 18 as compared to EMT6 xenografts. This is a luminal and not basal (CK5 or CK18) keratin (Fig. 1h). However when EMT6 cells are incubated with mMDSC derived from 4T1 tumors, these upregulate CK14 and undergoes expansion of the CD24+CD29+ CSC-like population. These points need clarification.

Reviewer #1 (Remarks to the Author):

This is interesting study describing the role of gMDSC in tumor metastases in mouse model of breast cancer. Most of the experiments are well designed and conclusions are convincing.

Major concern is with rather large amount of data that are mostly confirmatory of already established in many publications facts. It includes detailed description of changes in MDSC in 4T1 model, the role of mMDSC in EMT (including STAT3 signaling), the role of gMDSC in promoting tumor metastases. There are new data concerning changes in tumor cells caused by gMDSC and elegant experiments in Figure 8.

We greatly appreciate the reviewer's positive comments on our work.

Authors stated that EMT signature was not associated with tumor prognosis and

referenced one paper (page 9). However, brief analysis of the literature showed more than two dozen papers reported just opposite conclusions. In this respect data in Figure 6 look rather incrementally different from already published data.

We agree with the reviewer that there are multiple previous reports suggesting either correlate with response to therapy or poor survival in patients with high EMT-related gene expression signature. However, our studies in breast cancer confirms the data by Soundararajan et al., who reported in 2015 that neither any EMT related gene alone or in combination of these genes predict poor clinical survival in breast cancer patients. More specifically, we examined whether EMT-related genes (IL1A, IL6, NOS2, Twist1, Twist2, Vim, TGFB1) will predict poor survival in breast cancer patients using the recent TCGA data set (published in Cell 2015). As shown below these genes did not predict poor survival.

Specific concerns

1. Authors did not established causal role of gMDSC in the observed phenomenon by depleting these cells;

We now provided evidence that depletion of gMDSCs in metastatic 4T1 tumor-bearing mice substantially reduced the pulmonary metastasis and that correlated with reduced gMDSC infiltration in lungs.

2. The authors check the cytokine secretion profile in co-culture of MDSC with tumor cells. It was not clear how they determined what cytokines were secreted only by tumor cells and not by MDSC (figure S2; they normalize data to secretion profile or tumor cells alone, but not info about secretion from MDSC alone).

When we performed these assays, we determined the value of cytokine production in co-cultures by subtracting the value of production by either tumor cells alone or MDSCs alone.

3. In co-culture the authors do not mention GM-CSF and there is no indication of MDSC viability (and/or phenotype) after the co-culture.

In our co-culture experiments, we cultured MDSC subsets in tumor cell media which is RPMI with 10% FBS and did not use the GM-CSF since it induces granulocytic differentiation. However, We demonstrated that after two days culture, majority of MDSCs were live and maintained their phenotype as shown in this figure.

4. The authors use immunocompetent mice (BalbC and B6), but they don't mention or discuss how they control the effect of endogenous neutrophils/MDSC during tumor

development and when they inject exogenous MDSC in TBM.

Although we agree with reviewers concern, our experimental design is set up with proper controls such as the mice with only tumors without MDSC injection demonstrated no sign of metastasis, if endogenous neutrophils/MDSCs had any effect on tumor then we would expect a difference between the control mice with only tumor injection and the mice with tumor and MDSC injection.

Minor comments

- 1. the authors should quantify luciferase signal in mice and show statistical significance (some mice have very low signal that could be just background), also few mice are shown, so it is important to understand what is the frequency of the observed results.*
- 2. the authors should show that after induction tumors keeps growing (luciferase detection and quantification)*
- 3. The differences in signaling (Figure 4m-n) aren't discussed, or validated (for example, inhibit the signal with exogenous inhibitors gives the same results??). Also total stat 1 and stat 3 are missing in the WB.*

In the revised manuscript, we adequately addressed those minor concerns.

Reviewer #2 (Remarks to the Author):

This is an elegant manuscript demonstrating a dynamic and reversible tumor cell plasticity involving epithelial to mesenchymal transition (EMT) and mesenchymal to epithelial transition (MET), which are hallmarks of metastatic dissemination and colonization. Whereas metastatic spread involves EMT leading to invasion, colonization at distant sites requires MET, a process conducive metastatic cell proliferation. Using co-culture experiments in vitro and in vivo, the authors demonstrate a critical interaction between mammary tumor cells and infiltrating monocytic and granulocytic subsets of myeloid derived suppressor cells (mMDSC, gMDSC) at the primary tumor and distant organs. They show that mMDSC stimulate EMT at primary tumor and that gMDSC

stimulate MET at the lungs. A “metastatic gene signature” was identified that was further shown to predict poor patient survival in several human cancers, suggesting clinical relevance of these studies.

This study uses multiple and alternative approaches to demonstrate the cross-communication between mMDSC and tumor cells in inducing EMT in tumors, leading to stemness and invasion. They also clearly show that infiltration of gMDSC in the lungs stimulate MET, which is conducive of tumor cell proliferation leading to lung colonization.

We appreciate the favorable comments of the reviewer on our manuscript.

There remains however several issue to be addressed as outlined below.

1. This study revolves around tumor infiltrated immune cells or myeloid cells that differentiate into cells that promote tumor growth and invasion in addition to their immunosuppressive role. They show that mMDSC and gMDSC are immunosuppressive, however, it is not clear how the immunosuppressive activity is contributing to the diverse metastatic inducing properties.

We agree with the reviewer that our studies suggest a link between immunosuppressive activity and tumor plasticity. One mechanism MDSCs exert their immunosuppressive activity is via NOS2 production which we demonstrated to have increased by more than 3000 fold in mMDSCs derived from 4T1 tumor bearing mice. Using NOS2 donor (DPTA) and inhibitor (1400W), we now demonstrated that NOS2 activation in tumor cells induces the production of inflammatory cytokines such IL1, IL6, TGFb and thus EMT phenotype. Furthermore, mMDSC induced expressions of these cytokines in tumor cells were partially suppressed when NOS2 was inhibited by 1400W suggesting that NOS2 may be critical link between immunosuppression and tumor plasticity.

2. “4T1 tumors showed pulmonary infiltrates as early as one week post implantation and

also displayed enlarged spleens". It is not clear which immune cells are causing spleen enlargement and how these are related to the immunosuppressive mMDSC or gMDSC (Fig. 1c).

We believe that enlargement of spleens in 4T1 tumor-bearing mice is due to high levels of gMDSC infiltration (45-50%) since spleen weight was reduced in gMDSC depleted mice.

3. While infiltration of mMDSC peaks after one week in 4T1 primary tumor following implantation, it continues to infiltrate the lungs up to 3 weeks (Fig. 2a) suggesting mMDSC may play a role in lung colonization, although the authors show that they do not in co-culture experiments with the non-metastatic EMT6 cells. Also, it appears that the infiltration of mMDSC in AT-3 tumors (Fig. 2g) does not follow the same kinetics as for 4T1. What is the basis for this discrepancy?

We agree with the reviewer's observation, however, this difference is due to the growth kinetics of 4T1 and AT-3 tumors in respective mice models. AT-3 tumors grow relatively slow and therefore they take little longer time to induce the accumulation of MDSCs in primary tumor and lung. We now have extended MDSC analyses up to 5 weeks in AT-3 tumor-bearing mice and determined that it has a similar pattern of MDSC infiltration.

4. Fig. 3g-h shows upregulation of vimentin and twist (EMT) upregulation in EMT6 co-cultured with mMDSC and upregulation of Ki67 (MET) in cells co-cultured with gMDSC. Is Twist expressed in vimentin positive cells and not in ki67+ cells?

Although we failed the immunofluorescent detection of Twist1 in formalin fixed tissues, we demonstrated by western blotting assay that mMDSCs induced the expression of Twist1.

5. Fig. 4 demonstrates that mMDSC infiltrating 4T1 tumors cause CSC expansion after co-culturing with EMT6 cells using the stem cell markers CD24+CD29+. Can this

population be isolated and tested for tumor initiating potential and metastatic potential?

We performed additional experiments and demonstrated that mMDSC induced CSC population with CD24⁺CD29⁺ phenotype were more tumorigenic than the CD24⁻CD29⁺ phenotype.

6. The authors show changes that indicate enrichment of CSCs as indicated above. They show increases in p-STAT and NFκB when tumor cells are incubated with mMDSC and increase in p-Erk when tumor cells are co-cultured with gMDSC (Fig. 4m-n). Although implicit in the EMT and MET phenotypes, these markers remain correlative. Is p-ERK expression more prominent at sites of gMDSC infiltration in the lungs and p-STAT at sites of EMT/invasion or at the leading edge. Is p-Erk associated with ki67+ cells (Fig 5a)?

We were able to demonstrate increased pStat1 expression in invasive tumor front where mMDSCs are localized. However, pErk staining was challenging since there was very high background that made it difficult to differentiate tumor cells from other surrounding cells.

7. 4T1 cells are thought to be similar to basal or triple negative breast cancer cells, yet 4T1 xenografts upregulate keratin 18 as compared to EMT6 xenografts. This is a luminal and not basal (CK5 or CK18) keratin (Fig. 1h). However when EMT6 cells are incubated with mMDSC derived from 4T1 tumors, these upregulate CK14 and undergoes expansion of the CD24⁺CD29⁺ CSC-like population. These points need clarification.

It was also surprising for us to find that Krt18 is one of the highly upregulated genes in 4T1 xenografts compared to EMT6 xenografts. As stated by the reviewer, 4T1 tumor cells represents human basal subtype and has no or very limited expression of luminal keratins such as CK18, however, we believe that when implanted orthotopically tumors cells may upregulate the expressions of some of these luminal keratins to be able to

grow in vivo in mice. Consistent with the literature (Weng et al, 2007, Molecular Cancer Research) that stimuli in the tumor microenvironment may induce the expression of CK18 in vivo. For the upregulation of CK14 in EMT6 tumor cells by mMDSCs derived from 4T1 tumor-bearing mice is consistent with previous publications such as Kevin et al., 2013, Cell. In revised manuscript, we commented on these findings.

Reviewers' comments:

Reviewer #1 (Remarks to the Author):

Authors made an effort to address my concerns. However, it was largely superficial.

My concern about novelty was largely ignored. However, I think data as presented could be considered novel enough.

Concern about co-culture was not address. It is still not clear if the difference in cytokine secretion in co-culture is due to different release from tumor or MDSC. Subtraction of the background is necessary, however, it does not tell whether cytokines produced by MDSC. Analysis of gene if not protein expression in tumors or MDSC after co culture should be performed.

Viability of Ly6G+ cells without cytokines is a concern. It was not clear why authors did not show Ly6G/Ly6C in same graphs, as well as expression of macrophage markers. As presented, the appearance of Ly6Glow macrophages in this culture could not be excluded.

All luciferase graphs (curves) are without SD, even if claimed in the legend. Also, not clear how many mice used in each group, and why only few are shown. Or how many time experiments were done.

- a. In fig 7 (b-k) only 2/5 mice show mets (i) and one in panel j.
- b. In figure 8 only 4 mice shown, and only half have tumors. Figure legend doesn't match data (mean and SD are not reported at all).

Reviewer #2 (Remarks to the Author):

The authors have performed a comprehensive and extensive sets of experiments to address the reviewers comments. The differential effects of mMDSCs and gMDSCs on metastatic dissemination via effects on EMT and MET, involving invasion versus proliferation are compelling. The identification of a "metastatic gene signature" that predicts poor survival in breast several other solid tumors is very important as EMT signature is not predictive of metastasis. The ability of conditioned media (CM) released by the metastatic 4T1 cells in stimulating metastasis of indolent tumor cells (Figure 7) is impressive. However, the manuscript lacks in dept in identifying or at least discussing the potential factors in the CM that might cause the recruitment of gMDSCs to the lungs. Overall the manuscript includes a critical mass of evidence to support the role of mMDSCs and gMDSCs in stimulating EMT and MET leading to metastasis. I support the publication of this manuscript.

Our point by point response to reviewers' critiques is as follows;

Reviewer #1 (Remarks to the Author):

Q: Authors made an effort to address my concerns. However, it was largely superficial. My concern about novelty was largely ignored. However, I think data as presented could be considered novel enough.

A: We greatly appreciate reviewer's constructive comments on our first version of the manuscript and we believe that addressing those comments strengthened our manuscript. Reviewer also acknowledges that the data presented in this revised manuscript is novel. The novelty of our study is to be the first to provide evidence of tumor plasticity which is regulated by two different subsets of MDSCs in primary site and in lungs.

Q: Concern about co-culture was not address. It is still not clear if the difference in cytokine secretion in co-culture is due to different release from tumor or MDSC. Subtraction of the background is necessary, however, it does not tell whether cytokines produced by MDSC. Analysis of gene if not protein expression in tumors or MDSC after co culture should be performed.

A: We certainly understand the concern of the reviewer and agree with him that once has to be very careful doing these kinds of studies to be certain as to what cell type is

producing the cytokines. However, we believe that we made every effort to be certain of our data that is presented in this manuscript. As explained in the manuscript, we simplified the data for the reader by subtracting the values of cytokines by either tumor or MDSCs alone. As shown below, we have included all possible controls such as tumor cells alone, MDSC subsets alone and the co-cultures of tumor cells and MDSC subsets. As demonstrated, some cytokines secreted by tumor cells at higher levels while other secreted by MDSC subsets at higher level. However, co-cultures of tumor cells with mMDSC (red color) or gMDSC (blue color) produced much higher levels of these cytokines. Since it is technically impossible to differentiate what cell type producing these cytokines, we performed qPCR analyses to verify some of these cytokines (IL1, IL6, G-CSF, TGFb CCL3, S100A8/A9 and other genes such as Vimentin and NOS2. We indeed verified these genes by qPCR in two different mouse models (BALB/c and C57BL/6J) already presented in our manuscript (Figure 4d,e,f,g and Figure 5g,h,i,j).

Q: Viability of Ly6G+ cells without cytokines is a concern. It was not clear why authors did not show Ly6G/Ly6C in same graphs, as well as expression of macrophage markers. As presented, the appearance of Ly6Glow macrophages in this culture could not be excluded.

A: Although we understand reviewer's concern of MDSC viability in *in vitro* culture, we can assure the reviewer that *in vitro* culturing of MDSCs for 24-48 hr did not result in massive cell death as shown below. As requested by the reviewer, we also included Ly6C/Ly6G flow analyses below. It is known that monocytic MDSCs may differentiate into granulocytes and macrophages when cultured *in vitro*. In addition, we have an MDSC expert in our team who has published extensively in this field. All the MDSC studies were approved by him.

Q: All luciferase graphs (curves) are without SD, even if claimed in the legend. Also, not clear how many mice used in each group, and why only few are shown. Or how many time experiments were done.

A: We now added SDs in graphs and indicated the number of mice used in each experiments in our revised manuscript.

Q: a. In fig 7 (b-k) only 2/5 mice show mets (i) and one in panel j.

b. In figure 8 only 4 mice shown, and only half have tumors. Figure legend doesn't match data (mean and SD are not reported at all).

A: Although reviewer claims only 2 mice showing lung mets in control groups (Fig 7i), in fact 4 out of 5 mice developed lung metastasis that show signal in live imaging of animals. We now included the ex-vivo imaging of lungs demonstrating a visible luciferase signals in 4 out of 5 mice in control group, whereas there is only one lung showing luciferase signal in ly6G antibody treated group.

Reviewer #2 (Remarks to the Author):

Q: The authors have performed a comprehensive and extensive sets of experiments to address the reviewers comments. The differential effects of mMDSCs and gMDSCs on

metastatic dissemination via effects on EMT and MET, involving invasion versus proliferation are compelling. The identification of a “metastatic gene signature” that predicts poor survival in breast several other solid tumors is very important as EMT signature is not predictive of metastasis. The ability of conditioned media (CM) released by the metastatic 4T1 cells in stimulating metastasis of indolent tumor cells (Figure 7) is impressive. However, the manuscript lacks in dept in identifying or at least discussing the potential factors in the CM that might cause the recruitment of gMDSCs to the lungs. Overall the manuscript includes a critical mass of evidence to support the role of mMDSCs and gMDSCs in stimulating EMT and MET leading to metastasis. I support the publication of this manuscript.

A: We are extremely delighted by reviewer’s favorable comments and appreciation of our manuscript. One potential factor for systemic MDSC induction and recruitment might be the extremely high levels of G-CSF expressed/secreted by metastatic tumor cells compared to the indolent tumors. However, gMDSC infiltration in lungs may be dependent on S100A8/S100A9 since we already demonstrated significantly higher levels of these chemokines in lung infiltrated gMDSCs and in tumor cells upon co-culture with gMDSCs. Consistent with our findings, Hiratsuko et al, (Nature Cell Biol, 2008), reported that S100A8/S100A9-induced serum amyloid A3 directly recruits MDSCs to lungs and facilitates metastasis in melanoma and lung cancers. We believe that detailed investigation of the mechanism of pulmonary MDSC infiltration will be out of the scope of our studies in this manuscript which aimed at understanding in vivo tumor plasticity and phenotypic switch during metastatic process. However, in our ongoing and future studies, we are certainly interested in determining the molecular and mechanistic recruitment of MDSCs in distant organs including lungs.

We greatly appreciate the thoughtful comments on our manuscript by the reviewers. We now believe that these modifications have significantly improved the manuscript and hope that is now suitable for publication in *Nature Communications*.

REVIEWERS' COMMENTS:

Reviewer #1 (Remarks to the Author):

Authors response is reasonable. I have no further major concerns.